# D₂O: Dynamic Discriminative Operations for Efficient Long-Context Inference of Large Language Models

**Zhongwei Wan**[1†*]  **Xinjian Wu**[2†]  **Yu Zhang**[3]  **Yi Xin**[4]  **Chaofan Tao**[5]
**Zhihong Zhu**[6]  **Xin Wang**[1]  **Siqi Luo**[4]  **Jing Xiong**[5]  **Longyue Wang**[7]  **Mi Zhang**[1]
[1]The Ohio State University  [2]University of Chinese Academy of Sciences  [3]Tongji University
[4]Nanjing University  [5]The University of Hong Kong  [6]Peking University
[7]Alibaba International Digital Commerce
`https://github.com/AIoT-MLSys-Lab/d2o`

## Abstract

Generative inference in Large Language Models (LLMs) is impeded by the growing memory demands of Key-Value (KV) cache, especially for longer sequences. Traditional KV cache eviction strategies, which discard less critical KV pairs based on attention scores, often degrade generation quality, leading to issues such as context loss or hallucinations. In this work, we introduce **D**ynamic **D**iscriminative **O**perations (**D₂O**), a KV cache compression method that optimizes KV cache size dynamically and discriminatively at two levels without fine-tuning, while preserving essential context. At **layer level**, D₂O leverages the varying densities of attention weights between shallow and deep layers to dynamically determine which layers should avoid excessive eviction via a novel *dynamic allocation strategy* to minimize information loss. At **token level**, D₂O incorporates a *compensation mechanism* that maintains a similarity threshold to re-discriminate the importance of currently discarded tokens, determining whether they should be recalled and merged with similar tokens. We conduct experiments on various benchmarks and LLM architectures. Our results show that D₂O not only achieves significant memory savings and enhances inference throughput by more than $3\times$ but also maintains high-quality long-text generation.

## 1 Introduction

Large Language Models (LLMs) (Achiam et al., 2023; Touvron et al., 2023; Meta, 2024; Jiang et al., 2023; Wan et al., 2023) excel in tasks requiring long contexts such as dialog systems (Chiang et al., 2023), document summarization (Zhang et al., 2023), question answering (Kamalloo et al., 2023), and code completion (Roziere et al., 2023). Such long contexts demand a significant amount of Key-Value (KV) cache. For instance, a model with 30 billion parameters, processing inputs of 1024 tokens at a batch size of 128, requires up to 180 GB KV cache (Zhang et al., 2024c). Such bottleneck underscores the critical need for KV cache optimization.

To minimize memory demands of KV cache, one effective method is KV cache eviction (Xiao et al., 2023b; Zhang et al., 2024c; Liu et al., 2023; Ren & Zhu, 2024; Zhang et al.; Ge et al., 2023), where the key is to identify a subset of KVs to be evicted from the cache. However, existing methods all suffer from both **layer-level** and **token-level** information loss. Specifically, at **layer level**, existing methods equally treat all the layers and indiscriminately evict KV pairs at each layer. However, *not all the layers exhibit the same patterns*. Figure 1 visualizes the attention weights on the GSM8K dataset (Cobbe et al., 2021a). As shown, the shallower layers (layer 0 and 1) display densely interconnected attention maps, while the deeper layers (layer 30 and 31) exhibit a staircase sparse pattern, where attention is localized to specific context segments, with only a few tokens in each segment receiving substantial attention. This observation aligns with findings from (Zhao et al., 2024a;b), indicating that while shallower layers primarily engage with syntactic structures through global attention, deeper layers target task-related semantic knowledge with localized attention. Consequently, applying the same eviction strategy indiscriminately across all the layers may compromise important information in long contexts. At **token level**, as shown in

---

*Project leader. †Equal contribution.

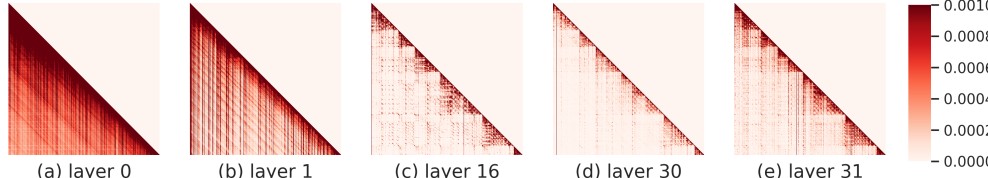

|       |       |       |       |       |
|-------|-------|-------|-------|-------|
| (a) layer 0 | (b) layer 1 | (c) layer 16 | (d) layer 30 | (e) layer 31 |

Figure 1: Attention map density comparisons of shallow layers (layer 0, 1) and deep layers (layer 16, 30, 31) of LLaMA-2-7B on the GSM8K dataset. We use the mean value of heads for each layer.

Figure 2 (a), existing methods enable models to operate within a constrained KV cache budget by either directly dropping KV pairs (e.g., StreamingLLM (Xiao et al., 2023b)) or selectively removing them based on specific eviction strategies, such as using cumulative attention scores (e.g., $H_2O$ (Zhang et al., 2024c)) or mean attention scores (e.g., RoCo (Ren & Zhu, 2024)). However, *the irreversible nature of eviction and the difficulty in accurately predicting which KV pairs are essential for future text generation can lead to information loss*, causing hallucinations, contextual inconsistencies (Yang et al., 2024b), and challenges in maintaining long-context integrity (Bai et al., 2024).

In this paper, we introduce **D**ynamic **D**iscriminative **O**perations (**$D_2O$**), a KV cache compression method that tackles the two fundamental issues of existing methods. The key idea of $D_2O$ is to incorporate dynamic discriminative operations at both **layer level** and **token level**. Specifically, at **layer level**, based on the findings in Figure 1, unlike existing methods that indiscriminately evict KV pairs, $D_2O$ proposes a *dynamic allocation strategy* using inverse variance softmax, adjusting the KV cache budget for each layer based on the density metric of the attention weights. At **token level**, given the uncertainties about how discarded tokens might affect future outputs, $D_2O$ introduces an effective *compensation mechanism* by maintaining an exponential moving average (EMA) threshold that assesses the degree of similarity between previously discarded and retained tokens, allowing $D_2O$ to dynamically decide whether a currently discarded token

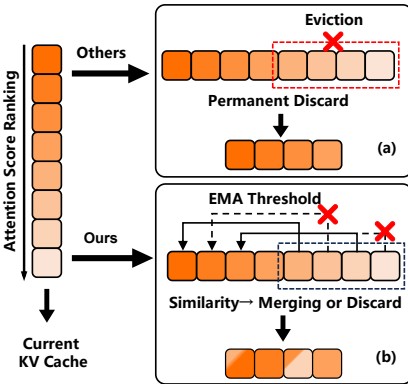

Figure 2: Existing methods (a) vs. $D_2O$ (b) at token level.

should be recalled and merged with a similar token retained in the current KV cache according to the current EMA threshold in order to compensate the information loss of KV cache eviction, as shown in Figure 2 (b). Through these operations at layer level and token level, $D_2O$ maintains the KV cache at a consistent size while being able to preserve valuable information from evicted tokens, enabling LLMs to handle the generation of extended texts with improved memory efficiency and high-throughput inference while minimizing the loss of contextual information.

We evaluate $D_2O$ and compare it with state-of-the-art KV cache compression methods: StreamingLLM (Xiao et al., 2023b), $H_2O$ (Zhang et al., 2024c), RoCo (Ren & Zhu, 2024), pyramidKV Cai et al. (2024b), and CaM (Zhang et al.). To demonstrate the generability of $D_2O$, we conduct our evaluation using four models from three different LLM families (Llama, Falcon, and Mistral) and diverse tasks involving math and commonsense reasoning, long-context QA, summarization, and code completion, drawn from LM-Eval (Gao et al., 2021), LongBench (Bai et al., 2023), long-context fact retrieval (Kamradt, 2023), and language modeling (Rae et al., 2019) benchmarks. We highlight five of our findings:

- (1) Under the same reduced KV cache budgets, $D_2O$ consistently achieves superior performance on reasoning tasks compared to state-of-the-art baselines.
- (2) For long-context tasks, $D_2O$ outperforms state-of-the-art baselines on LongBench with minimal accuracy degradation.
- (3) $D_2O$ also outperforms state-of-the-art baselines on the Needle In A Haystack task, demonstrating superior long-context fact retrieval capabilities.
- (4) Even with a limited KV cache, $D_2O$ is able to effectively leverage long-distance dependencies in language modeling, achieving superior performance compared to state-of-the-art baselines.
- (5) Lastly, $D_2O$ enables larger batch sizes and is able to achieve up to 3.04 times higher throughput than the original model in our experimental setup.

## 2 RELATED WORK

**Non-Trainable KV Cache Compression.** Majority of existing non-trainable KV cache compression methods reduce KV caches by focusing on evicting less crucial KVs. For example, Mistral-7B (Jiang et al., 2023), StreamingLLM (Xiao et al., 2023b) and SirLLMs (Yao et al., 2024) focus on tokens crucial for near-sequence generation. $H_2O$ (Zhang et al., 2024c), Scissorhands (Liu et al., 2023) and RoCo (Ren & Zhu, 2024) maintain a small set of influential tokens based on attention scores. FlexGen (Ge et al., 2023) adopts importance policies based on attention scores for KV eviction, while SnapKV (Li et al., 2024) uses a recent window strategy to compress KV cache for long prompts. PyramidKV (Cai et al., 2024b) and PyramidInfer (Yang et al., 2024a) propose the layer-wise KVs eviction strategies. NaCL (Chen et al., 2024) combines proxy-tokens and random eviction to keep robustness of LLMs. However, these methods can significantly lose context from evicted KVs.

**Trainable KV Cache Compression.** Some methods attempt to adapt LLMs to learn KV cache compression by training on selected datasets. For example, LESS (Dong et al., 2024) learns the residuals between the original and approximated attention outputs from a sparse policy applied during training. DMC (Nawrot et al., 2024) pre-trains on original data to learn parameters that control compression across various heads and layers in the KV cache. However, training on selected datasets poses challenges in adapting these methods to diverse downstream tasks due to limited generalizability. Unlike these approaches, $D_2O$ employs a plug-and-play method that requires no additional training, offering broader applicability without the need for dataset-specific tuning.

**Token Merging.** Token merging (Bolya et al., 2022; Shi et al., 2023) consolidates tokens into fewer, more meaningful units while preserving information integrity. This approach has emerged as a preferred alternative to token pruning for reducing the token count. Methods such as ToMe (Bolya et al., 2022), TPS (Wei et al., 2023), MG-ViT (Zhang et al., 2024b), and PPT (Wu et al., 2023) have applied token merging and pruning techniques for visual representation. However, these approaches mainly focus on merging the hidden states and are primarily designed for visual classification tasks. Recently, CaM (Zhang et al.) proposes to use a Bernoulli process to generate a mask for value state merging on the KV cache during long-text generation. It still employs a uniform merging strategy across all layers, disregarding variations in attention density between layers. $D_2O$ addresses the above issues by performing merging directly on the KV cache with a dynamic layer-level KV cache allocation and a dynamic merging strategy based on an EMA threshold. This prevents excessive information loss during KV cache compression in long-text generation tasks and improves efficiency for autoregressive tasks in LLMs.

## 3 PRELIMINARY: GENERATIVE INFERENCE WITH KV CACHE

Standard generative inference of an LLM includes two stages: prompt encoding and token generation.

**Prompt Encoding**. In the prompt encoding stage, a prompt sequence is utilized to generate a KV cache for each Transformer layer within an LLM. Consider an input prompt tensor $\mathbf{X} \in \mathbb{R}^{L_{\text{prompt}} \times D}$, where $L_{\text{prompt}}$ represents the length of the prompt and $D$ denotes the hidden dimension of the model. For simplicity, the indices for heads and layers are omitted. The key and value tensors are derived as:

$$\mathbf{K} = \mathbf{X}\mathbf{W}_K, \mathbf{V} = \mathbf{X}\mathbf{W}_V, \tag{1}$$

where $\mathbf{W}_K, \mathbf{W}_V \in \mathbb{R}^{D \times D}$ represents the weights for the key and value layers, respectively. Once $\boldsymbol{K}$ and $\boldsymbol{V}$ are computed, they are stored in the KV cache to facilitate the token generation process (Ott et al., 2019; Wolf et al., 2020).

**Token Generation**. In the token generation stage, KV cache is both utilized and updated to sequentially produce tokens. For each time step $i$, only the keys and values for the new token $\mathbf{x}_i$ are computed whereas those for $\mathbf{x}_{<i}$ are retrieved from the cache. Then the cache is updated and the output of newly generated token is as:

$$\mathbf{K} = [\mathbf{K}, \mathbf{x}_i\mathbf{W}_K], \mathbf{V} = [\mathbf{V}, \mathbf{x}_i\mathbf{W}_V], \tag{2}$$

$$\mathbf{x}_{i,out} = \text{Softmax}\left(\mathbf{q}_i\mathbf{K}^\top/\sqrt{D}\right)\mathbf{V}, \mathbf{q}_i = \mathbf{x}_i\mathbf{W}_Q, \tag{3}$$

where $\mathbf{W}_Q \in \mathbb{R}^{D \times D}$ is the weight matrix of the query layer. The linear expansion of KV cache with each new token significantly increases memory usage and latency, particularly for longer prompts, underscoring the importance of compressing the KV cache.

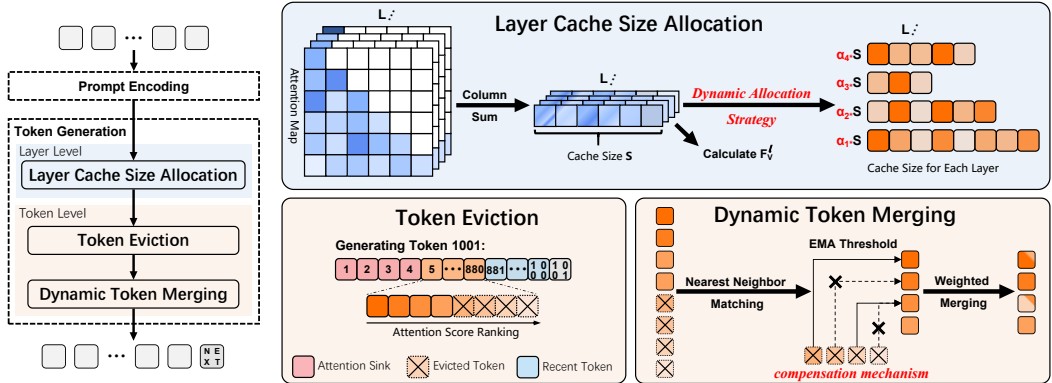

Figure 3: Overview of D$_2$O. For the **layer cache size allocation** at the layer level, D$_2$O addresses the issue of inconsistent attention density across higher and lower layers by incorporating a dynamic cache at each layer. The size of the cache is determined by the variance metric of attention and *dynamic allocation strategy*. At the token level, D$_2$O addresses long-context information loss by incorporating a combination of a **token eviction** scheme and a **dynamic token merging** technique (where *compensation mechanism* is located).

## 4    D$_2$O

Figure 3 provides an overview of D$_2$O. For simplicity, each layer in the figure is depicted with a single attention head. The dynamic **layer-level** and **token-level** discriminative operations proposed by D$_2$O are explained in Section 4.1 and Section 4.2, respectively in details.

### 4.1    DYNAMIC LAYER-LEVEL DISCRIMINATIVE OPERATION

Employing a uniform layer-wise size, such as those proposed in H$_2$O (Zhang et al., 2024c) and StreamingLLM (Xiao et al., 2023b), across all layers could potentially compromise model performance. To address this, we propose using a specific metric, $F_v^l$, to evaluate the attention density of each layer $l$:

$$F_v^l = \mathrm{Var}\left(\sum_{i=0}^{L_{\mathrm{prompt}}} \mathbf{A}_p^l[i,:]\right), \quad \mathbf{A}_p^l = \mathrm{Softmax}\left(\mathbf{Q}_p^l \mathbf{K}_p^{l\top}/\sqrt{D}\right), \qquad (4)$$

where $\mathbf{A}_p^l$ denotes the attention score of prompt encoding in each layer, and $\mathbf{Q}_p^l, \mathbf{K}_p^l \in \mathbb{R}^{L_{\mathrm{prompt}} \times D}$. We sum the elements of each column in $\mathbf{A}_p^l$ to establish the initial state of the cumulative attention sequence. The attention density for each layer is then quantified by the variance of this sequence, as denser attention weights correspond to smaller variances. This relationship is illustrated in Figure 4. We observe a consistent phenomenon across all models on the GSM8K dataset: the variance of attention scores is lower in the shallow layers (e.g., 0, 1, 2) and middle layers (e.g., 13, 14), indicating that the attention weights are dense, as also shown in Figure 1. This density makes it difficult to distinguish which tokens should be discarded. In deeper layers, the variance increases and the attention weights display a sparser pattern.

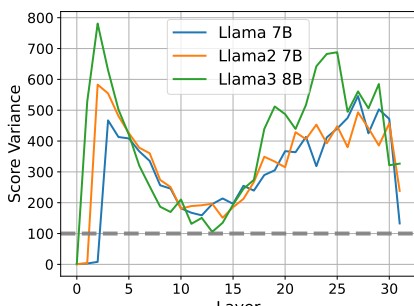

Figure 4: Variances of attention score across different layers for various models.

**Dynamic Allocation Strategy**. Leveraging this consistency, we propose a new *dynamic allocation strategy* using inverse variance softmax to adjust the KV cache size in each layer: layers with higher variance $F_v^l$ are allocated a smaller cache size, while shallower layers with lower variance receive a larger cache allocation. Specifically, for a given compression ratio $\rho$, the cache size for each layer $S_l$, is calculated as:

$$S_l = \alpha_l \cdot S, \quad \text{where } \alpha_l = \frac{\exp(-F_v^l)}{\sum_{l=1}^{L} \exp(-F_v^l)} \cdot L \cdot \rho, \qquad (5)$$

where $\rho$ represents the compression ratio of the cache size, and $S$ is the original cache size, which is equal to $L_{\mathrm{prompt}}$ in the prompt encoding stage, and $L$ denotes the number of model layers. We adopt a softmax-like function to dynamically distribute cache proportions $\alpha_l$. The derivation details and theoretical analysis can be found in Appendix A.2 and Appendix A.9, respectively.

## 4.2 DYNAMIC TOKEN-LEVEL DISCRIMINATIVE OPERATION

After the layer-level discriminative operation, to compensate for the loss of long-context information, we introduce two critical strategies in the token-level discriminative operation: token eviction and dynamic token merging strategies. Specifically, while $D_2O$ is compatible with any token eviction technique, we propose an eviction strategy based on accumulative attention (Zhang et al., 2024c) to dynamically prune KV cache in generation tasks. For token merging, we introduce a new strategy that utilizes a similarity threshold based on exponential moving average (EMA) to dynamically determine whether to merge discarded tokens back into the preserved KV cache through weighted merging.

### 4.2.1 TOKEN EVICTION

The core concept of token eviction involves dynamically updating the KV cache by leveraging cumulative attention scores. This process systematically excludes the least essential KV pairs to maintain the compressed cache size $S_l$ for each layer, thereby preserving only the most valuable tokens for efficient inference. A recent study (Xiao et al., 2023b) suggests that retaining crucial attention sink tokens within the most recent KV cache window enhances the stability of attention score distributions across extended texts. Unlike traditional accumulation-based approaches such as $H_2O$ (Zhang et al., 2024c), our strategy improves performance by maintaining attention sink tokens from the initial $T$ tokens of the input and integrating them within a recent window of size $M$. The attention score is formulated as follows:

$$\text{AttnScore} = \begin{cases} \sum_{i=0}^{L_{\text{prompt}}} \mathbf{A}_p[i,:], & \text{if token } i <= L_{\text{prompt}}, \\ \text{Softmax}\left(\mathbf{q}_i \mathbf{K}^\top / \sqrt{D}\right) + \sum_{i=1}^{L_{\text{prompt}}} \mathbf{A}_p[i,:], & \text{otherwise, token generation} \end{cases} \quad (6)$$

After obtaining the current cumulative attention scores, we retain the most recent window of size $M$ and include $T$ attention sink tokens. We then select the top $N$ tokens with the highest scores from the remaining KV cache to finalize the eviction process. The procedure is defined as follows:

$$\mathbf{K}_c = [\mathbf{K}[:T,:], \mathbf{K}[I,:], \mathbf{K}[-M:,:]], \quad \mathbf{V}_c = [\mathbf{V}[:T,:], \mathbf{V}[I,:], \mathbf{V}[-M:,:]], \quad (7)$$
$$\text{and } I = \text{Top}_N\left(\text{AttnScore}[T:-M], N\right), \quad (8)$$

where $\text{Top}_N(\cdot, N)$ selects the top $N$ important tokens with indices $I$ in AttnScore, $(\mathbf{K}_c, \mathbf{V}_c)$ represents the preserved KV cache after eviction, and $S = T + N + M$ denotes the current cache size.

### 4.2.2 DYNAMIC TOKEN MERGING

Directly discarding the eviction tokens (i.e., $\mathbf{K}_e = \mathbf{K} - \mathbf{K}_c$) may compromise the integrity of the long context. To mitigate information loss, we propose a dynamic token merging approach that retrieves tokens still containing potential value at minimal computational cost and integrates these selected tokens with similar reserved tokens. Considering the alignment properties of KV pairs, we compute the similarity matrix only on the key tokens and share both the similarity metric and weighted merging weights with the value tokens. We outline this approach in three key steps: nearest-neighbor matching, EMA threshold judgment, and weighted merging.

**Nearest Neighbor Matching**. We utilize a many-to-one nearest-neighbor matching algorithm (Dang et al., 2021) to compute the similarity matrix $\mathbf{U}$ between $\mathbf{K}_e$ and $\mathbf{K}_c$. We then identify the most similar tokens from $\mathbf{K}_c$ as candidates for merging. Specifically, let $I^e$ and $I^c$ denote the indices, and $L^e$ and $L^c$ represent the lengths of tokens in $\mathbf{K}_e$ and $\mathbf{K}_c$, respectively. Each element $u_{i,j}$ in $\mathbf{U}$ represents the interaction between tokens for matching, where $i \in I^e$ and $j \in I^c$. We then determine the closest token $\mathbf{k}_*^{\text{nearest}}$ in $\mathbf{K}_c$ for each evicted token $\mathbf{k}_i$. The formulas are as follows:

$$\mathbf{k}_*^{\text{nearest}} = \underset{j \in I^c}{\text{Argmax}}\left(u_{i,j}\right), \text{ where } u_{i,j} = \frac{\mathbf{k}_i^\top \mathbf{k}_j}{\|\mathbf{k}_i\| \|\mathbf{k}_j\|} \quad (9)$$

Here, we adopt cosine similarity, where $\|\cdot\|$ denotes the norm. Since the similarity matrix $\mathbf{U} \in \mathbb{R}^{L^e \times L^c}$ is derived directly from input prompts and $\mathbf{U} \in \mathbb{R}^{L^c}$ during token generation, it introduces no additional parameters and ensures computational efficiency.

**EMA Threshold**. After calculating similarities and identifying candidate tokens $\mathbf{K}_*^{\text{nearest}}$, directly applying average weighted fusion to token pairs could lead to feature dispersion (Liang et al., 2022). Additionally, as depicted in Figure 1, attention patterns in higher layers exhibit a staircase-like structure, indicating a focus on local window information. Moreover, since only a few critical tokens exist outside these local windows, indiscriminately merging all candidate tokens can introduce redundant information or noise, ultimately affecting inference accuracy. Inspired by the exponential moving average (EMA) (Hunter, 1986; Busbridge et al., 2023) used in time-series tasks—which prioritizes more recent data and enhances sensitivity to data changes—we propose an EMA threshold for token-level operations. Our approach emphasizes the importance of recent similarity between current evicted tokens and preserved tokens while smoothing historical similarity information between previously evicted and preserved tokens. Specifically, the influence of past token similarity thresholds diminishes exponentially over time, assigning greater weight to more recent thresholds. The EMA threshold is formulated as:

$$\tau_t = \begin{cases} \frac{1}{L^e} \sum_{i=0}^{L^e} \text{Max}(\mathbf{U}_t[i,:]), & \text{if } t = 0 \text{ for prompt encoding} <= L_{\text{prompt}}, \mathbf{U}_t \in \mathbb{R}^{L^e \times L^c} \\ \beta \text{Max}(\mathbf{U}_t[:]) + (1-\beta)\, \tau_{t-1} & \text{otherwise}, t > 0 \text{ for token generation}, \mathbf{U}_t \in \mathbb{R}^{L^c} \end{cases} \tag{10}$$

The initial threshold $\tau$ is set to the average of the highest similarity values between evicted tokens and the conserved set, as computed from the similarity matrix $\mathbf{U}_t$ at each forward step $t$. The smoothing constant $\beta$ regulates the balance between the current similarity matrix $\mathbf{U}_t$ and the previous similarity threshold $\tau_{t-1}$, with higher values of $\beta$ increasing sensitivity to changes in current similarity. If the maximum similarity of a given evicted token is lower than $\tau_i$, it is permanently discarded. Otherwise, a weighted merging strategy is applied.

**Weighted Merging**. Lastly, for a conserved token, evicted tokens with higher similarity should be assigned greater weights. Therefore, we use weighted merging instead of averaged merging, as this approach mitigates potential errors arising from imperfect token scoring. The weighted merging formulas are defined as:

$$\mathbf{k}_{cj} = \mathbf{w}_{cj}\mathbf{k}_{cj} + \sum_{\mathbf{k}_{ei} \in \mathbf{K}_e} \mathbf{w}_{ei}\mathbf{k}_e, \quad \mathbf{v}_{cj} = \mathbf{w}_{cj}\mathbf{v}_{cj} + \sum_{\mathbf{v}_{ei} \in \mathbf{V}_e} \mathbf{w}_{ei}\mathbf{v}_e, \tag{11}$$

where $\mathbf{w}_c$ and $\mathbf{w}_e$ represent the weights assigned to each preserved and evicted key-value pair, respectively. We adopt a similarity-based weighting strategy, inspired by Graph Attention Networks (GAT) (Veličković et al., 2017). The weight calculation is as follows:

$$\mathbf{w}_{cj} = \frac{e}{\sum_{\mathbf{k}_{ei} \in \mathbf{K}_e} \exp(\mathbf{u}_{ij})\mathbf{m}_{ij} + e}, \quad \mathbf{w}_{ei} = \frac{\sum_{\mathbf{k}_{ei} \in \mathbf{K}_e} \exp(\mathbf{u}_{ij})\mathbf{m}_{ij}}{\sum_{\mathbf{k}_{ei} \in \mathbf{K}_e} \exp(\mathbf{u}_{ij})\mathbf{m}_{ij} + e}, \tag{12}$$

where $\mathbf{m}_{i,j} \in \mathbf{M}$ represents the mask matrix of $\mathbf{U}$. If $\mathbf{x}_j \in \mathbf{K}_c$ is the most similar token to $\mathbf{x}_i \in \mathbf{K}_e$, then $\mathbf{m}_{i,j} = 1$; otherwise, $\mathbf{m}_{i,j} = 0$. The fusion weights $\mathbf{w}_{cj}$ and $\mathbf{w}_{ei}$ are determined by the mask values $\mathbf{m}_{i,j}$ and similarities $\mathbf{u}_{i,j}$. Specifically, $\mathbf{w}_{ei}$ represents the weight for each evicted token $\mathbf{k}_{ei}$, while $\mathbf{w}_{cj}$ corresponds to the weight of the preserved token itself. Each conserved token $\mathbf{k}_{cj}$ retains the highest fusion weight, as its self-similarity equals one. Therefore, preserved tokens remain unchanged, as the most similar tokens are not modified, whereas evicted tokens are integrated into their most similar counterparts, replacing the originals.

## 5 EXPERIMENTS AND ANALYSIS

### 5.1 EXPERIMENTAL SETUP

**Models.** We evaluate D$_2$O using four models: Llama-2 (Touvron et al., 2023), Llama-3 (Meta, 2024), Falcon (Almazrouei et al., 2023), and Mistral (Jiang et al., 2023). For Llama-2, we employ model sizes ranging from 7B to 13B. For Llama-3, we use the 8B model. Notably, both Llama-2 and Mistral utilize multi-head attention, whereas Falcon employs multi-query attention (Shazeer, 2019), and Llama-3 adopts grouped-query attention (Ainslie et al., 2023). We implement D$_2$O using the Hugging Face Transformers codebase (Wolf et al., 2019).

**Tasks.** We evaluate D$_2$O using datasets with both standard and extended context lengths. For standard contexts, we utilize generation tasks from LM-Eval (Gao et al., 2021), assessing model performance across commonsense and math reasoning on CoQA (Exact Match (EM) Accuracy) (Reddy

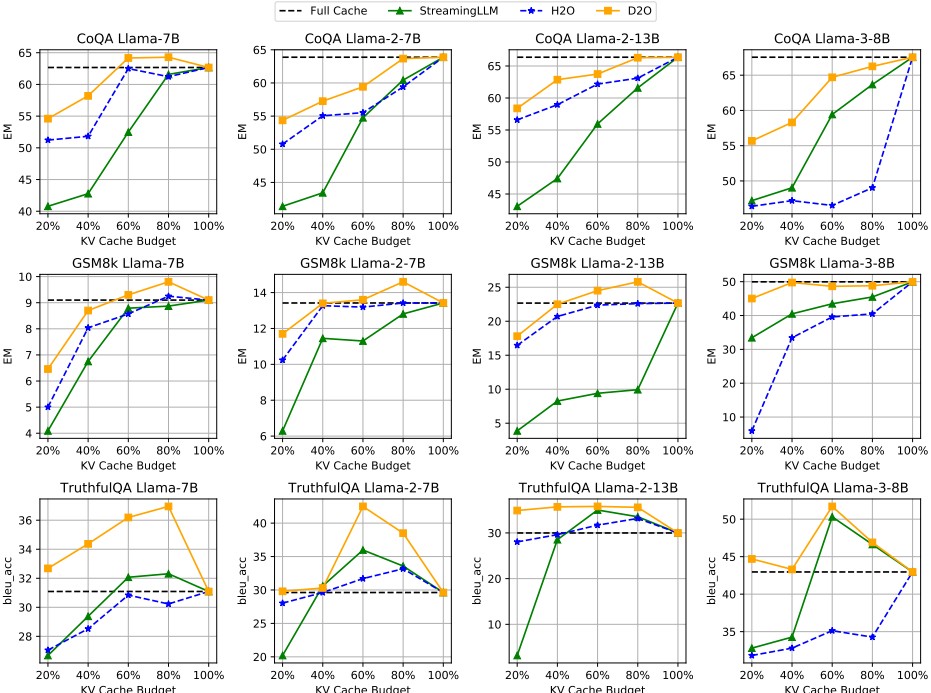

Figure 5: Performance of $D_2O$ and other methods for LLama backbones on reasoning datasets including CoQA, GSM8K, and TruthfulQA.

et al., 2019), TruthfulQA (BLEU score) (Lin et al., 2022), and GSM8K (Exact Match (EM) Accuracy) Cobbe et al. (2021b). For long-context tasks, we apply LongBench (Bai et al., 2023), which is particularly suited for evaluating the effects of compressed KV cache. This involves tasks from subgroups such as Single-Document QA, Multi-Document QA, Summarization, Synthetic, and Code Completion. Additionally, we assess $D_2O$'s capability in long-context retrieval with the Needle-in-a-Haystack test (Briakou et al., 2023), challenging the model to retrieve a specific sentence within a large document. We also validate our method's long sequence modeling ability using PG-19 (Rae et al., 2019). More details are illustrated in Appendix A.1.

**Implementation.** In our principal experiment, we select the $\beta$ value for EMA threshold merging from the range $0.5 \sim 0.9$. The hyperparameter experiment in Appendix A.3 shows that $\beta = 0.7$ achieves optimal performance. Additionally, the number of top $N$ important tokens and the recent token $M$ are typically set as $N = 3 : 1$, with $N + M + T = \alpha_l \cdot S$. The overall KV cache compression ratio $\rho$ varies across different settings, including values such as 0.2, 0.4, and 0.8. All experiments were conducted on NVIDIA A100 80GB GPUs. Further details on implementation and the determination of hyperparameters are provided in Appendix A.1 and Appendix A.3, respectively.

## 5.2 COMPARATIVE ANALYSIS UNDER DIFFERENT KV CACHE COMPRESSION RATIOS

In Figure 5, we benchmark $D_2O$ on GSM8K, CoQA, and TruthfulQA datasets.We compare models equipped with a full KV cache against those utilizing our $D_2O$ compression technique over several Llama models: Llama-1-7B, Llama-2-7B, Llama-2-13B, and the latest Llama-3-8B. The ratio represents the proportion of the overall compressed KV cache budget relative to the prompt length $L_{prompt}$. Results indicate that $D_2O$ consistently outperforms all other KV compression methods across all configurations. Notably, $D_2O$ significantly enhances performance, particularly under reduced budgets conditions. This performance highlights $D_2O$'s ability to retain crucial context, compensating for the severe loss of information inherent in eviction-based methods and preventing the degradation of LLMs' reasoning capabilities, particularly for Llama-3-8B on the GSM8K and CoQA datasets. Intriguingly, on the TruthfulQA dataset, $D_2O$ even surpasses the full KV cache model across four Llama backbone settings and most of the budget ratios, demonstrating that $D_2O$'s unified eviction and dynamic merging strategies can prune irrelevant tokens from LLMs' input text, preserving essential context and then enhancing reasoning accuracy.

Table 1: Performance evaluation of $D_2O$ on various models in LongBench benchmarks. For each baseline, except for the full model, we retain 20% of $L_{prompt}$ ($\rho = 0.2$, $S = L_{prompt}$) as the preserved KV cache size and highlight the best method.

| | Methods | Single-Document QA | | | Multi-Document QA | | | Summarization | | | Summarization | | | Synthetic | | Code | |
| | | NrtvQA | Qasper | MF-en | HotpotQA | 2WikiMQA | Musique | GovReport | QMSum | MultiNews | TREC | TriviaQA | SAMSum | PCount | PRe | Lcc | RB-P |
|---|---|---|---|---|---|---|---|---|---|---|---|---|---|---|---|---|---|
| **Falcon-7B** | Full Model | 1.03 | 3.82 | 7.62 | 1.75 | 1.78 | 1.25 | 3.77 | 2.48 | 6.04 | 5.00 | 8.27 | 2.73 | 0.49 | 0.41 | 9.98 | 7.33 |
| | Local Window | 0.43 | 2.60 | 4.80 | 1.26 | 1.31 | 0.50 | 1.79 | 0.61 | 2.31 | 4.67 | 7.00 | 1.72 | 0.84 | 0.39 | 8.95 | 7.28 |
| | StreamingLLM | 0.32 | 2.61 | 4.85 | 1.56 | 1.04 | 0.54 | 1.80 | 0.53 | 4.54 | 4.67 | 6.53 | 1.68 | 1.60 | 0.36 | 9.67 | 7.16 |
| | H2O | 0.43 | 4.01 | 5.65 | 1.66 | 1.70 | 0.50 | 2.61 | 0.61 | 4.36 | 4.33 | 7.57 | 2.20 | 0.53 | 0.0 | 10.57 | 6.72 |
| | RoCo | 0.38 | 3.82 | 5.33 | 1.31 | 1.61 | 0.38 | 2.25 | 0.53 | 3.88 | 4.35 | 7.42 | 1.88 | 0.49 | 0.22 | 9.78 | 6.75 |
| | CaM | 0.58 | 3.80 | 5.77 | 1.79 | 1.65 | 0.65 | 2.68 | 0.59 | 4.13 | 4.57 | 7.38 | 2.41 | 0.66 | 0.23 | 10.35 | 6.85 |
| | D2O | **0.94** | **4.27** | **6.50** | **1.72** | **2.16** | **0.79** | **3.01** | **3.22** | **4.75** | **5.32** | **8.36** | **2.47** | **2.05** | **0.85** | **11.25** | **7.32** |
| **Mistral-7B** | Full Model | 26.28 | 29.8 | 49.44 | 41.77 | 26.52 | 19.35 | 33.32 | 24.44 | 26.28 | 66.67 | 86.16 | 41.11 | 4.43 | 90.5 | 56.91 | 49.09 |
| | Local Window | 16.25 | 15.72 | 29.25 | 27.88 | 19.55 | 12.80 | 21.64 | 15.71 | 15.45 | 33.65 | 26.54 | 18.57 | 2.35 | 41.25 | 28.50 | 26.50 |
| | StreamingLLM | 18.75 | 16.22 | 33.54 | 29.77 | 19.42 | 13.34 | 18.55 | 17.78 | 17.54 | 50.52 | 62.76 | 20.88 | 2.39 | 45.22 | 52.31 | 33.28 |
| | H2O | 22.45 | 23.52 | 42.78 | 33.56 | 23.45 | 15.58 | 28.48 | 18.88 | 20.22 | 56.72 | 75.52 | 32.88 | 3.45 | 78.55 | 52.38 | 37.25 |
| | RoCo | 19.55 | 21.22 | 38.54 | 29.88 | 19.98 | 13.38 | 25.22 | 15.32 | 16.85 | 52.45 | 76.23 | 30.50 | 2.88 | 75.58 | 49.54 | 38.75 |
| | CaM | 22.47 | 23.40 | 42.64 | 33.83 | 23.02 | 15.90 | 28.36 | 18.99 | 19.82 | 56.25 | 75.28 | 32.62 | 3.39 | 78.79 | 52.68 | 36.85 |
| | D2O | **24.54** | **25.72** | **45.07** | **34.84** | **24.92** | **17.29** | **29.70** | **21.90** | **24.06** | **62.99** | **84.02** | **38.03** | **4.18** | **86.26** | **55.17** | **46.15** |
| **Llama-2-7B** | Full Model | 15.02 | 8.92 | 21.89 | 9.12 | 10.2 | 3.71 | 19.45 | 21.29 | 1.42 | 61.00 | 89.81 | 39.73 | 2.49 | 4.94 | 67.95 | 55.14 |
| | Local Window | 3.27 | 6.56 | 2.3 | 8.88 | 7.29 | 1.25 | 0.06 | 2.07 | 0.28 | 17.67 | 4.55 | 4.70 | 1.44 | 5.88 | 17.69 | 13.81 |
| | StreamingLLM | 10.31 | 5.62 | 19.75 | 6.65 | 8.75 | 2.49 | 1.29 | 19.86 | 1.32 | 52.67 | 88.96 | 37.13 | 0.59 | 6.10 | 64.76 | 50.49 |
| | H2O | 14.31 | 7.15 | 20.45 | 8.61 | 9.93 | 3.29 | 9.96 | 20.22 | 0.40 | 59.67 | 88.46 | 39.61 | 2.31 | 7.75 | 65.00 | 53.40 |
| | RoCo | 12.22 | 6.58 | 18.45 | 7.76 | 7.95 | 3.52 | 8.88 | 19.56 | 0.55 | 57.65 | 85.54 | 36.14 | 2.55 | 4.84 | 61.59 | 50.55 |
| | CaM | 11.31 | 6.24 | 18.95 | 8.06 | 8.44 | 3.91 | 9.35 | 19.40 | 1.46 | 57.83 | 85.19 | 36.61 | 3.38 | 4.98 | 61.73 | 49.94 |
| | D2O | **16.69** | **7.88** | **21.45** | **9.26** | **10.58** | **4.06** | **16.18** | **21.37** | **1.41** | **59.82** | **89.70** | **40.43** | **3.86** | **7.09** | **66.56** | **53.82** |
| **Llama-2-13B** | Full Model | 12.91 | 9.37 | 19.65 | 11.19 | 10.84 | 5.59 | 19.39 | 21.37 | 4.74 | 63.33 | 87.37 | 42.3 | 4.67 | 7.92 | 67.36 | 54.62 |
| | Local Window | 3.77 | 5.17 | 2.78 | 13.83 | 11.76 | 3.98 | 0.14 | 1.48 | 0.32 | 17.67 | 7.54 | 3.63 | 0.67 | 3.89 | 18.44 | 13.64 |
| | StreamingLLM | 7.19 | 5.70 | 11.62 | **14.06** | 10.20 | 4.51 | 2.28 | 17.91 | 0.39 | 52.00 | 85.25 | 37.64 | 2.17 | 5.00 | 64.05 | 46.34 |
| | H2O | 13.52 | 6.53 | 15.10 | 10.74 | 10.74 | 5.28 | 12.13 | 20.48 | 0.29 | 60.33 | 85.73 | 42.23 | 3.25 | 9.52 | 64.98 | 51.31 |
| | RoCo | 11.01 | 4.88 | 14.05 | 10.22 | 9.88 | 4.95 | 9.54 | 19.85 | 1.07 | 55.56 | 84.78 | 38.95 | 3.22 | 6.02 | 63.21 | 51.95 |
| | CaM | 11.17 | 5.59 | 13.98 | 10.64 | 10.72 | 4.81 | 9.40 | 20.59 | 2.01 | 56.04 | 85.07 | 39.20 | 3.61 | 6.27 | 63.36 | 52.11 |
| | D2O | **14.66** | **8.09** | **16.59** | 10.83 | **12.41** | **5.88** | **16.13** | **21.16** | **3.36** | **62.57** | **88.15** | **42.75** | **6.07** | **9.83** | **67.19** | **52.81** |
| **Llama-3-8B** | Full Model | 14.25 | 12.89 | 22.45 | 11.03 | 12.17 | 6.98 | 30.80 | 23.25 | 4.02 | 71.00 | 90.10 | 42.08 | 6.33 | 12.51 | 72.94 | 61.26 |
| | Local Window | 1.78 | 4.64 | 4.10 | 6.11 | 6.91 | 2.81 | 0.56 | 10.33 | 0.02 | 33.5 | 28.67 | 10.56 | 5.69 | 2.00 | 32.80 | 23.68 |
| | StreamingLLM | 10.47 | 9.96 | 13.82 | 9.64 | 11.05 | 5.53 | 19.99 | 20.53 | 3.13 | 62.67 | 90.05 | 41.30 | 5.44 | 14.05 | 70.44 | 57.93 |
| | H2O | 13.27 | 11.05 | 17.72 | 10.38 | 11.23 | 6.38 | 21.29 | 21.33 | 3.38 | 66.63 | 89.19 | 41.12 | 5.52 | 11.11 | 71.86 | 58.29 |
| | RoCo | 10.77 | 10.55 | 16.54 | 9.98 | 8.95 | 9.52 | 20.78 | 20.15 | 2.59 | 63.98 | 86.26 | 38.59 | 5.55 | 10.05 | 68.78 | 56.66 |
| | CaM | 11.15 | 11.02 | 16.84 | 10.47 | 8.83 | 9.45 | 21.23 | 20.73 | 2.57 | 64.10 | 87.21 | 38.69 | 5.86 | 10.40 | 69.72 | 57.51 |
| | D2O | **14.43** | **12.66** | **19.93** | **11.92** | **12.79** | **9.88** | **24.36** | **23.42** | **3.95** | **69.72** | **90.99** | **42.36** | **6.61** | **14.67** | **72.43** | **60.00** |

## 5.3 ACCURACY COMPARISON ON LONG-CONTEXT TASKS

**LongBench Results.** We evaluate $D_2O$ on five models using LongBench, as shown in Table 1, including Falcon-7B, Mistral-7B, Llama-2-7B, Llama-2-13B, and Llama-3-8B. To assess the performance of $D_2O$ and various baselines under high compression conditions, we set the default KV cache budget ratio to $\rho = 0.2$. Table 1 demonstrates that $D_2O$ effectively manages KV cache compression with minimal impact on accuracy and successfully captures key information in lengthy texts compared to the full model. In particular, the local window method suffers from severe performance degradation due to significant context loss. Furthermore, we compare $D_2O$ with other recent eviction-based baselines to further demonstrate its capability to retain key information. The results show that $D_2O$ significantly outperforms other eviction-based methods, such as StreamingLLM (Xiao et al., 2023a), H$_2$O (Zhang et al., 2024c), RoCo (Ren & Zhu, 2024), and CaM (Zhang et al.), especially on the Llama-3-8B backbone. In addition to the experiments involving 7B, 8B, and 13B models presented in Table 1, we conducted additional experiments with the Code-Llama-34B model, as shown in Table 2. We evaluated the 34B model on the LongBench code task type. The results indicate that $D_2O$ outperforms other eviction-based baselines and closely matches the performance of the full KV cache model. This demonstrates that $D_2O$ effectively generalizes to larger LLM scales, validating its scalability and robustness across different model sizes.

Table 2: LongBench code performance on 34B model.

| Code-LLama-34B | Lcc | RB-P |
|---|---|---|
| Full | 75.52 | 65.46 |
| Local | 37.38 | 21.49 |
| StreamLLM | 63.42 | 55.27 |
| H2O | 68.85 | 61.02 |
| RoCo | 67.21 | 59.21 |
| CaM | 69.21 | 58.71 |
| D2O | **75.28** | **64.77** |

**Long-Context Fact Retrieval Task.** To validate $D_2O$'s retrieval capabilities in long contexts after compressing the KV cache, we employ the Needle In A Haystack task (Kamradt, 2023), designed to retrieve specific "needles" from extensive documents. We adopt the evaluation settings from the Retrieval Head study, using Llama-2-7B-80k as the backbone of the experiment.

For a fair comparison, the KV cache budget was set to 4096 and 8192, and both $D_2O$ and the baseline models were tested on contexts with maximum lengths of 50k and 100k. The average accuracy is reported in Table 3. $D_2O$ not only outperforms other eviction-based methods but also exhibits the smallest drop in performance accuracy when compared to the full model even without the assistance of a well-designed retriever, especially when the cache budget is 8192.

Table 3: Needle-in-a-haystack results.

| Methods | L=50k | L=100k | L=50k | L=100k |
|---|---|---|---|---|
| Full Model | 97.88 | 94.46 | 97.88 | 94.46 |
| | 4096 | | 8192 | |
| StreamingLLM | 58.64 | 47.93 | 62.84 | 51.34 |
| $H_2O$ | 79.84 | 69.81 | 82.32 | 72.34 |
| SnapKV | 83.55 | 76.22 | 86.63 | 80.42 |
| CaM | 82.66 | 78.22 | 87.59 | 78.88 |
| $D_2O$ | **91.27** | **87.74** | **94.48** | **91.88** |

These results underscore $D_2O$'s robust long-context retrieval capabilities, even with a compressed KV cache.

**Long Sequence Modeling Perplexity.** We sample data from PG-19 (Rae et al., 2019) to evaluate long-sequence language modeling perplexity. To ensure a fair comparison, we set the KV cache capacity to 2048. Figure 6 depicts the cumulative average negative log-likelihood (NLL) as a function of context length. $D_2O$ enables LLMs to process long sequences more effectively, achieving superior performance (lower perplexity) compared to other eviction-based compression methods. These results demonstrate that $D_2O$ can effectively leverage long-distance dependencies in language modeling, even with a limited KV cache.

Figure 6: Long sequence modeling PPL.

## 5.4 DYNAMIC ALLOCATION POLICY ANALYSIS

To evaluate the impact of our proposed dynamic allocation policy at the layer level, we conduct an experiment comparing various designs for the cache allocation factor $\alpha_l$, where each design influences how the cache is distributed across

Table 4: Comparison of cache allocation strategies.

| Method | CoQA | TREC |
|---|---|---|
| Exponential Decay | 48.26 | 57.46 |
| Uniform Allocation | 55.30 | 65.10 |
| Reciprocal of Variance | 58.10 | 68.00 |
| Inverse Variance Softmax (Ours) | **59.25** | **69.72** |

layers based on their attention variance $F_v^l$. We consider several settings: **(1)** *Inverse Variance Softmax (Ours)*: $\alpha_l = \frac{\exp(-F_v^l)}{\sum_{l=1}^{L} \exp(-F_v^l)} \cdot L \cdot \rho$, which allocates smaller cache sizes to layers with higher variance; **(2)** *Reciprocal of Variance*: $\alpha_l = \frac{1}{F_v^l} \cdot \frac{1}{\sum_{l=1}^{L} \frac{1}{F_v^l}} \cdot L \cdot \rho$, where the cache allocation is inversely proportional to the attention variance; **(3)** *Exponential Decay Allocation*: $\alpha_l = e^{-F_v^l}$, where cache is allocated in an exponentially decreasing manner based on the variance. **(4)** *Uniform Allocation*: $\alpha_l = \rho$, where all layers are assigned equal cache sizes. As shown in Table 4, the comparison of these settings demonstrates the effectiveness of our proposed approach in efficiently distributing cache based on variance, leading to superior performance in key metrics.

## 5.5 THROUGHPUT ANALYSIS

We demonstrate that reducing the KV cache with $D_2O$ significantly enhances real-world throughput, as illustrated in Table 5. All experiments are conducted using the Llama-3-8B architecture on an A100-80G GPU without CPU offloading. The KV cache budget is set equal to the length of the prompts to maintain the contextual in-

Table 5: Throughput comparison of full model and $D_2O$. 32 (256) means the max batch size is 32 with a 256 cache budget.

| Prompt+Gen | 256+1024 | 512+2048 | 1024+4096 | 2048+8192 |
|---|---|---|---|---|
| | Max Batch Size | | | |
| Full Model | 8 | 4 | 2 | 1 |
| $H_2O$ | 32 (256) | 16 (512) | 8 (1024) | 4 (2048) |
| $D_2O$ | 32 (256) | 16 (512) | 8 (1024) | 4 (2048) |
| | Throughput: tokens /s | | | |
| Full Model | 374.79 | 198.94 | 96.95 | 43.44 |
| $H_2O$ | 919.77 (2.45×) | 511.75 (2.57×) | 281.36 (2.90×) | 134.66 (3.10×) |
| $D_2O$ | 878.62 (2.34×) | 495.50 (2.49×) | 272.28 (2.80×) | 132.45 (3.04×) |

tegrity of the input prompts. We observe that $D_2O$ reduces memory usage, enabling larger batch sizes and higher throughput. Specifically, as text length increases, $D_2O$'s throughput advantage over the full model also grows. For example, throughput improves from 2.34× at the 256+1024 setting to 3.04× at the 2048+8192 setting, demonstrating $D_2O$'s efficiency in processing longer texts. Moreover, the table above also shows that our method achieves throughput comparable to $H_2O$ for long sequence inference, with a throughput 3.0-3.1 times that of the Full Model. This validates the GPU memory

efficiency of our merging strategy in increasing throughput compared to the full cache size. While our merging strategy incurs some computational overhead compared to $H_2O$'s eviction-based strategy, the benefits are clear in long text generation and inference tasks. As demonstrated in Table 1 (Longbench Results) and Appendix A.4 (Multi-turn Conversations) in the main text, $D_2O$ significantly reduces information loss due to eviction, enhancing inference accuracy. Furthermore, additional details on computational cost analysis are provided in Appendix A.5.

## 5.6 ABLATION ANALYSIS

In this section, we conduct a series of experiments to investigate the importance of each component and parameter setting in our proposed method. Unless otherwise specified, Llama-3-8B is used as the default model under the cache budget 20%.

**Ablation Study of Each Component.** To demonstrate the effectiveness of each component, we have included a table comparing the Full Model, $H_2O$, $D_2O$, and its key components in Table 6. We evaluated these models on four datasets from Longbench: Single-Document QA (MF-en), Multi-Document QA (2WikiMQA), Summarization (GovReport), and Synthetic (PRe). As shown in the following table, the removal of any component of $D_2O$ results in performance degradation. When two components are removed (leaving only the weighted merging), the performance loss is even greater, but still better than the $H_2O$ method, which is purely eviction-based. These results demonstrate that each component of $D_2O$ effectively mitigates the information loss associated with eviction-based KV optimization.

Table 6: Performance of each component.

|  | MF-en | 2WikiMQA | GovReport | PRe |
|---|---|---|---|---|
| Full Model | 22.45 | 12.17 | 30.8 | 12.51 |
| $H_2O$ | 17.22 | 11.23 | 21.29 | 11.11 |
| $D_2O$ | **19.93** | **12.79** | **24.36** | **14.67** |
| *w.o.* Layer Operation | 18.89 | 11.67 | 22.45 | 13.44 |
| *w.o.* EMA-Threshold | 19.23 | 11.88 | 22.85 | 13.23 |
| *w.o.* Both | 17.85 | 11.58 | 21.87 | 11.98 |

Table 7: Feature choice.

| Feature | CoQA | TREC |
|---|---|---|
| $\mathcal{K}$ | **59.25** | **69.72** |
| $\mathcal{V}$ | 53.88 | 64.34 |
| $\mathcal{K}/\mathcal{V}$ | 53.54 | 62.76 |

Table 8: Performance comparison with different merge policy.

| Methods | CoQA | TREC |
|---|---|---|
| Average | 56.58 | 67.12 |
| Weighted average | **59.25** | **69.72** |

Table 9: Ratio impact.

| Methods | CoQA | TREC |
|---|---|---|
| 1:1 | 56.52 | 66.45 |
| 1:3 | 55.43 | 65.41 |
| 3:1 | **59.25** | **69.72** |

**Token Similarity Metric.** We examine various choices for token similarity metrics based on keys, values, or both, to determine which tokens should be merged. As shown in Table 7, attention keys ($\mathcal{K}$) exhibit significantly higher performance than attention values ($\mathcal{V}$). We also observe a notable decrease in performance when using independent metrics for the key-value ($\mathcal{K}/\mathcal{V}$) cache, which we attribute to the disruption of the corresponding relationships between key-value pairs within the cache.

**Merge Policy.** After determining which tokens to merge, we explore the merging policy in our proposed method. Specifically, we compare the performance of average merging and weighted merging. The results in Table 8 indicate that the weighted merging policy achieves superior performance.

**Balancing Important Token Size ($N$) and Recent Size ($M$).** We also investigate the impact of different important token sizes and recent size ratios on performance, given a fixed budget. This ratio determines the emphasis placed on influential tokens from historical contexts (larger ratio) versus tokens from recent contexts (smaller ratio). The results in Table 9 suggest that important tokens reflecting global information have a greater impact on performance.

## 6 CONCLUSION

In this paper, we propose Dynamic Discriminative Operations ($D_2O$) that can effectively address the challenges of KV cache management in LLMs by dynamically merging tokens to maintain essential contextual information without requiring fine-tuning. By leveraging the varying densities of attention features across layers, $D_2O$ minimizes information loss during eviction and significantly reduces both computational and memory demands. Our experiments confirm that $D_2O$ not only preserves the quality of generation in long-text scenarios but also achieves an optimal balance between KV-cache compression and performance. Future research could explore integrating $D_2O$ with additional compression methods like quantization, distillation, and efficient attention architectures.

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

# A  APPENDIX

## A.1  MORE SETTING DETAILS

In all our experiments, we used model weights downloaded from Huggingface as follows: for all Llama architectures, the Llama-1-7B model employed the 'huggyllama/llama-7b'[1] checkpoint, Llama-2-7B used the 'meta-llama/Llama-2-7b-hf'[2] version, Llama-2-13B utilized 'meta-llama/Llama-2-13b-hf'[3], and for the latest Llama-3-8B, we used 'meta-llama/Meta-Llama-3-8B'[4]. In the Mistral architecture, the 'mistralai/Mistral-7B-Instruct-v0.2'[5] checkpoint was employed. For the Falcon architecture, 'tiiuae/falcon-7b'[6] was used. Additionally, for the evaluation metrics of the various sub-tasks such as "narrativeqa," "qasper," "multifieldqa_en," and "hotpotqa" within the LongBench benchmark, please refer to the official benchmark repository[7].

## A.2  DERIVATION OF CACHE ALLOCATION FACTOR $\alpha_l$

In this section, we provide the derivation for the cache allocation factor $\alpha_l$, which dynamically adjusts the KV cache size based on the variance of attention scores in each layer.

We begin by noting that the total cache size across all layers is constrained by the original fixed cache size $S$ for each layer and the compression ratio $\rho$. The total cache size after compression is given by:

$$\sum_{l=1}^{L} S_l = S \cdot L \cdot \rho \tag{13}$$

where $S_l$ is the cache size allocated to layer $l$, $L$ is the total number of layers, and $\rho$ represents the compression ratio. To allocate cache dynamically, we introduce $\alpha_l$, which governs the proportion of the total cache assigned to each layer. The cache size for layer $l$ is therefore:

$$S_l = \alpha_l \cdot S \tag{14}$$

and $\alpha_l$ must satisfy the constraint:

$$\sum_{l=1}^{L} \alpha_l = L \cdot \rho \tag{15}$$

To ensure layers with higher variance $F_v^l$ receive smaller cache sizes, we propose an inverse relationship between $\alpha_l$ and $F_v^l$. A softmax-like function is adopted to distribute cache proportions as follows:

$$\alpha_l = \frac{\exp(-F_v^l)}{\sum_{l=1}^{L} \exp(-F_v^l)} \cdot L \cdot \rho \tag{16}$$

This formulation ensures that layers with higher attention variance are allocated less cache, while those with lower variance receive more. The normalization factor $\sum_{l=1}^{L} \exp(-F_v^l)$ guarantees that the total allocation across all layers satisfies $\sum_{l=1}^{L} \alpha_l = L \cdot \rho$. Thus, the above formulas concludes the derivation of the dynamic cache allocation factor, where $\alpha_l$ is inversely proportional to the attention variance of each layer.

## A.3  MORE DETAILS OF HYPER-PARAMETERS DETERMINATION

Our hyper-parameters are designed to be both generalizable and robust across all tasks in the paper. We select hyper-parameters by first conducting hyper-parameter searches on specific long-text datasets from Longbench (e.g., TREC) and reasoning datasets from LM-Eval (e.g., COQA), as detailed in Section 5.6 (Ablation Study) of our paper. We then use the best-performing parameters for global

---

[1]https://huggingface.co/huggyllama/llama-7b

[2]https://huggingface.co/meta-llama/Llama-2-7b-hf

[3]https://huggingface.co/meta-llama/Llama-2-13b-hf

[4]https://huggingface.co/meta-llama/Meta-Llama-3-8B

[5]https://huggingface.co/mistralai/Mistral-7B-Instruct-v0.2

[6]https://huggingface.co/tiiuae/falcon-7b

[7]https://github.com/THUDM/LongBench

experiments. For token-level hyper-parameters such as $\beta$, we conducted searches within the range of 0.5-0.9 {0.5, 0.6, 0.7, 0.8, 0.9}. For $N$ (important tokens) and $M$ (recent tokens), we tested ratios {3:1, 2:1, 1:1, 1:2, 1:3}.

In addition to the TREC and COQA datasets used in the main text, we have conducted additional hyper-parameter ablation studies on the GSM8K (mathematical reasoning) and TruthfulQA (commonsense reasoning) datasets, as shown in Table 10 and 11. We found that the ratio of N:M =3:1 yielded the best performance across most datasets, as stated in Section 5.1 of our paper. For the EMA threshold parameter Beta, we observed that a value around 0.7 produced optimal results in most datasets, and we set the default Beta to 0.7.

| N:M | COQA | GSM8K | TruthfulQA |
|-----|------|-------|------------|
| 3:1 | **57.92** | **41.24** | **44.92** |
| 2:1 | 57.68 | 39.95 | 41.49 |
| 1:1 | 56.87 | 37.26 | 26.72 |
| 1:2 | 54.90 | 36.32 | 36.47 |
| 1:3 | 54.58 | 36.79 | 34.52 |

| $\beta$ | COQA | GSM8K | TruthfulQA |
|-----|------|-------|------------|
| 0.5 | 56.35 | 40.04 | 44.13 |
| 0.6 | 57.12 | 41.02 | **44.92** |
| 0.7 | **57.92** | **41.24** | 43.1 |
| 0.8 | 57.18 | 40.18 | 42.4 |
| 0.9 | 56.92 | 39.89 | 38.8 |

Table 10: N:M Comparison        Table 11: Beta Comparison

## A.4 GENERATED SAMPLES OF MULTI-TURN CONVERSATIONS

To validate our $D_2O$ method's ability to preserve critical context information and generate correct and fluent responses in multi-turn dialogues, we employed the MT-bench dataset (Zheng et al., 2024). This dataset consists of 3.3K expert-level pairwise human preferences for responses generated by six models, including GPT-4 and LLaMA-13B, in response to 80 multi-turn questions. It is specifically designed to assess the performance of language models in producing contextually appropriate conversations. To ensure a fair experimental comparison, we followed the settings (Xiao et al., 2023b; Zhang et al., 2024c) of using a KV cache budget of 2048 tokens. For $D_2O$ and $H_2O$, we set the quantity of the top $N$ important tokens at 48 and recent tokens $M$ at 2000.

Notably, due to the extremely long texts in streaming multi-turn dialogues, the full model will encounter out-of-memory issue. Therefore, we primarily compare $D_2O$ with $H_2O$ (Zhang et al., 2024c) and StreamingLLM (Xiao et al., 2023b). As illustrated in Figure 8, we randomly sample outputs according to the running order of the MT bench dialogue data, with samples 1 and 2 appearing in the earlier dialogue data and samples 3 and 4 in the latter part. From the outputs, we observe that during the early stages of multi-turn conversations, both $D_2O$ and two other eviction-based KV cache compression methods effectively captured the context and yielded accurate responses. However, after the second sample, $H_2O$ and StreamingLLM start to produce irrelevant content, losing conversational coherence. This deterioration underscores a significant loss of contextual information and a decline in future generation performance, illustrating the drawbacks of methods that directly drop the middle KV cache, such as StreamingLLM, or employ eviction strategies based on attention scores, like $H_2O$. Meanwhile, we observe that Full Cache encounters an out-of-memory error when processing the third sample. It highlights the critical necessity of optimizing KV cache to retain complete contextual information during multi-turn dialogues with ultra-long texts. This experiment demonstrates the efficacy of our two-level discriminative operations for context retention.

## A.5 COMPUTATIONAL COST ANALYSIS

To better understand the time overhead associated with prompt encoding and token generation after implementing token-level discriminative operations, we compared the time costs of prompt encoding to the total inference time. Our results, as illustrated in Table 12, show that the prompt encoding process, which utilizes token eviction and dynamic token merging operations, constitutes only a small fraction of the total time. Moreover, as the length of generation tokens increases, this proportion continues to decrease to 0.224%, indicating that the token merging operation is both efficient and less time-consuming.

Additionally, we compare the computational overhead of our method ($D_2O$) with eviction-based method ($H_2O$) in Table 13, we observe that the inference time for $D_2O$ is not significantly different from that of $H_2O$. This indicates that the token similarity calculation and the merging strategy are lightweight and do not affect the overall inference efficiency.

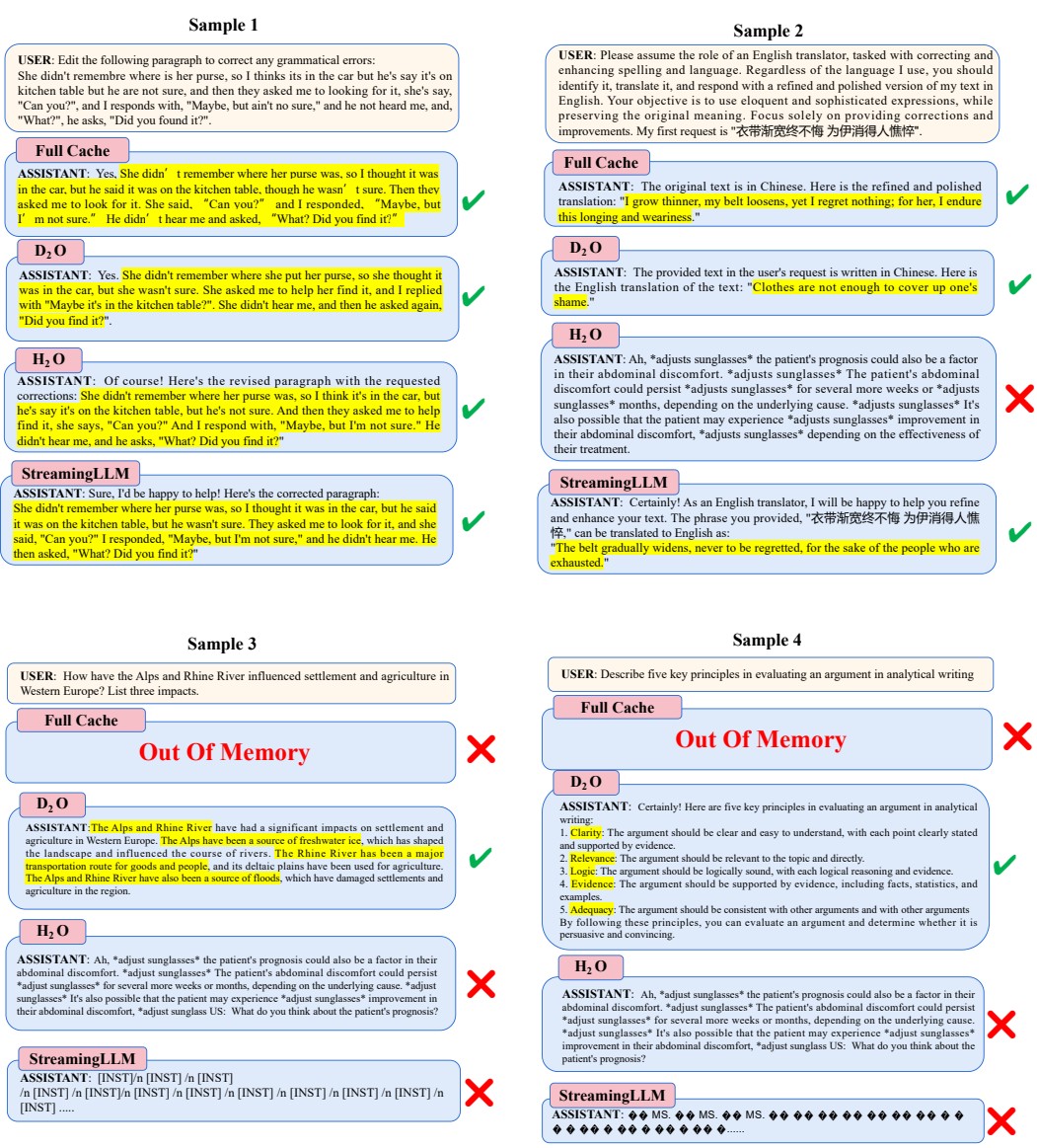

Figure 7: Overall caption for the two images.

Figure 8: A comparative visualization of text generation by the D$_2$O, H$_2$O, StreamingLLMs, and Full Cache methods is presented, with samples 1 to 4 collected sequentially according to the multiple rounds of dialogue from the MT Bench dataset. All three methods were tested on the Llama-2-7b-chat-hf model. The correct responses have been highlighted.

We also included additional experiments comparing latency across different model sizes, specifically Meta-Llama-3-8B, Llama-2-13B, Code-Llama-34B, and Meta-Llama-3-70B, under varying sequence lengths for encoding/prompt encoding. For these tests, we used a single A100 80G GPU for the 8B and 13B models, two GPUs for the 34B model, and four GPUs for the 70B model. As shown in the results in Table 14, inference time scales with both sequence length and model size, with larger models experiencing increased latency due to computational complexity and communication overhead, even with a fixed-size KV cache optimization.

Furthermore, to assess inference time and generation quality during long-generation tasks, we conducted additional tests on the longbook_sum_eng dataset from InfiniteBench (Zhang et al., 2024a), which has an average output length of 1.1K tokens. Due to computational constraints, we tested the first 20 samples using the Llama-3.1-8B-Instruct (128K) model, comparing $D_2O$ with representative baselines, including the eviction-based method ($H_2O$), layer-wise KV cache reduction (PyramidKV), and value token merge (CaM). The experiment was conducted using four A100 GPUs with 80GB. As shown in Table 15, $D_2O$ achieved the best performance, with a total inference time between $H_2O$ and PyramidKV and minimal latency differences.

For Table 16 of the Needle-in-a-haystack experiment, using a four-A100 80GB setup, our inference speed falls between SnapKV and CaM, while achieving the best performance among the methods.

Table 12: Inference time cost analysis of Llama 3-8B. The overall generation duration is calculated from the beginning of the decoding process to the conclusion of the generation sequence. Prompt encoding time spans from the initial prompt input to the completion of token eviction and dynamic token merging by $D_2O$. The KV cache budget is established at 256 tokens with a ratio of $M : N$ set at 1:3.

| Prompt Len + Decoding Len | Overall Generation Duration (s) | Prompt Encoding Duration (s) | Decoding Time Per Token (s) | Prompt Encoding/Overall (%) |
|---|---|---|---|---|
| 256+512 | 29.454 | 0.235 | 0.057 | 0.798% |
| 512+1024 | 58.528 | 0.246 | 0.057 | 0.420% |
| 1024+2048 | 121.191 | 0.328 | 0.059 | 0.271% |
| 2048+4096 | 232.398 | 0.520 | 0.057 | 0.224% |

Table 13: Computational overhead comparison (Cache size = 256, Reported by LLama 3-8B).

| Prompt Len + Decoding Len | Overall Generation Duration (s) | Prompt Encoding Duration (s) | Decoding Time Per Token (s) | Prompt Encoding/Overall (%) |
|---|---|---|---|---|
| $H_2O$ | | | | |
| 512+1024 | 52.253 | 0.214 | 0.051 | 0.410% |
| 2048+4096 | 214.425 | 0.315 | 0.523 | 0.153% |
| $D_2O$ | | | | |
| 512+1024 | 58.475 | 0.245 | 0.057 | 0.419% |
| 2048+4096 | 232.225 | 0.518 | 0.057 | 0.223% |

Table 14: Computational overhead analysis by model size and context length.

| Overall Generation Duration (s) | | | | |
|---|---|---|---|---|
| Prompt Len + Decoding Len | 8B | 13B | 34B | 70B |
| 256+512 | 29.454 | 53.02 | 147.27 | 279.81 |
| 512+1024 | 58.528 | 105.35 | 292.64 | 556.02 |
| 1024+2048 | 121.191 | 218.14 | 585.36 | 1151.31 |
| 2048+4096 | 232.398 | 418.32 | 998.79 | 2015.78 |

## A.6 Extended Analysis of LongBench Experiment

This experiment mainly validate the capability of our $D_2O$ to handle longer text data under a low KV cache budget, we selected several representative tasks from the LongBench, including single-document QA (e.g., MultifieldQA), multi-document QA (such as HotpotQA and 2wikimQA),

Table 15: Lookbook_sum_eng (Llama-3.1-8B-Instruct ) / (20% KV Cache)

| Llama-3.1-8B-Instruct (128K) | Rouge_Lsum (f1) | Inference time (min) |
|---|---|---|
| Full | 36.87 | 259.64 |
| $H_2O$ | 28.12 | 190.91 |
| PyramidKV | 31.92 | 206.06 |
| CaM | 33.82 | 181.51 |
| $D_2O$ | 35.48 | 201.27 |

Table 16: Needle-in-a-haystack results with inference time.

| Methods | L=50k | L=100k | L=50k | L=100k |
|---|---|---|---|---|
| Full Model | 97.88 (93.6 min) | 94.46 (175.4 min) | 97.88 (93.6 min) | 94.46 (175.4 min) |
| | 4096 | | 8192 | |
| StreamingLLM | 58.64 / 53.49 min | 47.93 / 100.23 min | 62.84 / 64.17 min | 51.34 / 110.92 min |
| $H_2O$ | 79.84 / 69.33 min | 69.81 / 125.93 min | 82.32 / 75.17 min | 72.34 / 138.88 min |
| SnapKV | 83.55 / 78.00 min | 76.22 / 146.17 min | 86.63 / 86.82 min | 80.42 / 154.34 min |
| CaM | 82.66 / 66.73 min | 78.22 / 116.24 min | 87.59 / 75.12 min | 78.88 / 129.80 min |
| $D_2O$ | 91.27 / 71.53 min | 87.74 / 129.85 min | 94.48 / 79.37 min | 91.88 / 142.22 min |

summarization (GovReport, TREC, and SAMSum), and code completion (Lcc RB-P). We specifically chose datasets exceeding 8k in length and only retained 20% KV cache budget. As shown in Table 17, $D_2O$ still demonstrates significant advantages even on datasets larger than 8k. Specifically, within the Llama-2-7B architecture, $D_2O$ outperforms the best baseline by 5.94 and 7.23 points in two summarization tasks, GovReport and TREC, respectively. This robustly validates the effectiveness of $D_2O$'s dynamic layer and token-level strategies, which effectively compress extended textual information under a low KV cache budget.

## A.7 Visualization of Long Context Fact Retrieval Task

We visualized the Needle-in-a-Haystack (Kamradt, 2023) test performance comparison of the full model, $H_2O$, and our $D_2O$ method to show the effectiveness of our dynamic token merging strategy for long-context information retrieval. As illustrated in Figure 9, we observed that the eviction-based method H2O, which relies on attention scores to prune the KV cache, loses significant contextual information, especially when the retrieval task reaches a maximum length of 100k. In contrast, our $D_2O$ method, which employs a dynamic token merging strategy, effectively preserves the information of evicted tokens and mitigates the impact of KV cache compression on long-context retrieval.

## A.8 Visualization of Attention Weights Across Various Datasets

In Figure 10, we visualize the attention weight results of the prompts across various layers of models such as Llama-1-7B and Llama-3-8B on reasoning datasets like GSM8K (Cobbe et al., 2021b) and COQA (Reddy et al., 2019). Consistent with the observations noted in Section 1 of the main text, a similar pattern exists across different models and datasets, wherein the lower layers of the models exhibit a higher density than the higher layers. Thus, this strongly corroborates our motivation for the layer-level discriminative operation, which employs different eviction ratio strategies for layers with varying densities of attention weights.

## A.9 Theoretical Analysis of the Dynamic Allocation Strategy

In this section, we provide a detailed theoretical analysis demonstrating why the proposed dynamic KV cache allocation strategy is superior to using a uniform compression ratio $\rho$ across all layers during the inference phase of LLMs. Our proof is grounded in information theory (Ash, 2012) and considers the multi-head attention mechanism inherent in Transformer-based models.

### A.9.1 Problem Formulation

Consider an LLM with $L$ layers, where each layer $l$ contains $H$ attention heads. During inference, each layer maintains its own key-value (KV) cache with a limited size due to resource constraints.

Table 17: Performance evaluation of $D_2O$ across various models using a range of benchmarks from LongBench at **8k settings**.

| Model | | MultifieldQA | HotpotQA | 2wikimQA | GovReport | TREC | SAMSum | Lcc | RB-P |
|---|---|---|---|---|---|---|---|---|---|
| | Full Model | 15.97 | 8.83 | 6.97 | 12.15 | 61.00 | 42.93 | 66.4 | 53.34 |
| **Llama-2-7B** | Local Window | 0.00 | 0.17 | 0.00 | 0.38 | 0.00 | 0.00 | 4.70 | 4.69 |
| | StreamingLLM | 15.05 | 6.68 | 5.77 | 6.72 | 52.67 | 41.39 | 62.17 | 46.82 |
| | $H_2O$ | 15.06 | 8.53 | 7.00 | **7.31** | 52.67 | 42.44 | 61.66 | 50.70 |
| | RoCo | 12.56 | 6.23 | 6.65 | 5.58 | 48.80 | 40.78 | 61.55 | 49.54 |
| | CaM | 12.46 | 6.44 | 7.02 | 5.61 | 49.07 | 41.18 | 61.46 | 49.71 |
| | **$D_2O$** | **16.58** | **9.89** | **8.68** | 14.07 | **60.00** | **44.75** | 65.88 | **54.31** |
| | ⇑ | 1.53 | 1.36 | 1.68 | 6.76 | 7.33 | 1.79 | 2.31 | 3.61 |
| | Full Model | 22.48 | 11.64 | 12.17 | 24.8 | 73.00 | 43.43 | 73.81 | 54.42 |
| **Llama-3-8B** | Local Window | 2.84 | 3.81 | 6.08 | 0.59 | 35.00 | 10.18 | 37.20 | 22.26 |
| | StreamingLLM | 12.93 | 9.25 | 8.70 | 19.20 | 67.00 | 39.40 | 71.99 | 52.08 |
| | $H_2O$ | 15.50 | 10.54 | 9.30 | 20.57 | 70.00 | 42.23 | 71.54 | 50.40 |
| | RoCo | 14.23 | 10.11 | 8.88 | 18.56 | 66.89 | 40.12 | 69.98 | 51.12 |
| | CaM | 14.04 | 10.57 | 8.78 | 18.97 | 67.29 | 40.00 | 70.04 | 50.98 |
| | **$D_2O$** | **18.57** | **12.50** | **10.43** | **22.72** | **71.96** | **44.64** | **73.95** | **54.57** |
| | ⇑ | 3.07 | 1.96 | 1.13 | 2.15 | 1.96 | 2.41 | 1.96 | 2.49 |

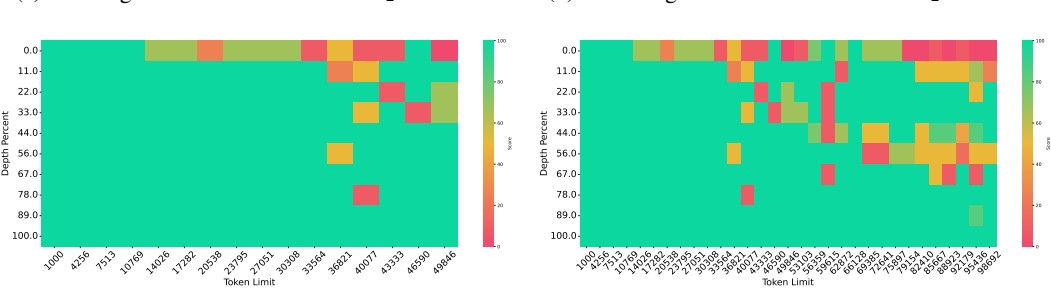

(a) 50k long-context fact retrieval of Full model.

(b) 100k long-context fact retrieval of Full model.

(c) 50k long-context fact retrieval of $H_2O$ method.

(d) 100k long-context fact retrieval of $H_2O$ method.

(e) 50k long-context fact retrieval of our $D_2O$.

(f) 100k long-context fact retrieval of our $D_2O$.

Figure 9: Visualization comparisons of long-context fact retrieval tasks for several methods.

The total cache size is constrained by:

$$\sum_{l=1}^{L}\sum_{h=1}^{H} S_l^h = S_{\text{total}} = \rho \cdot L \cdot H \cdot S, \tag{17}$$

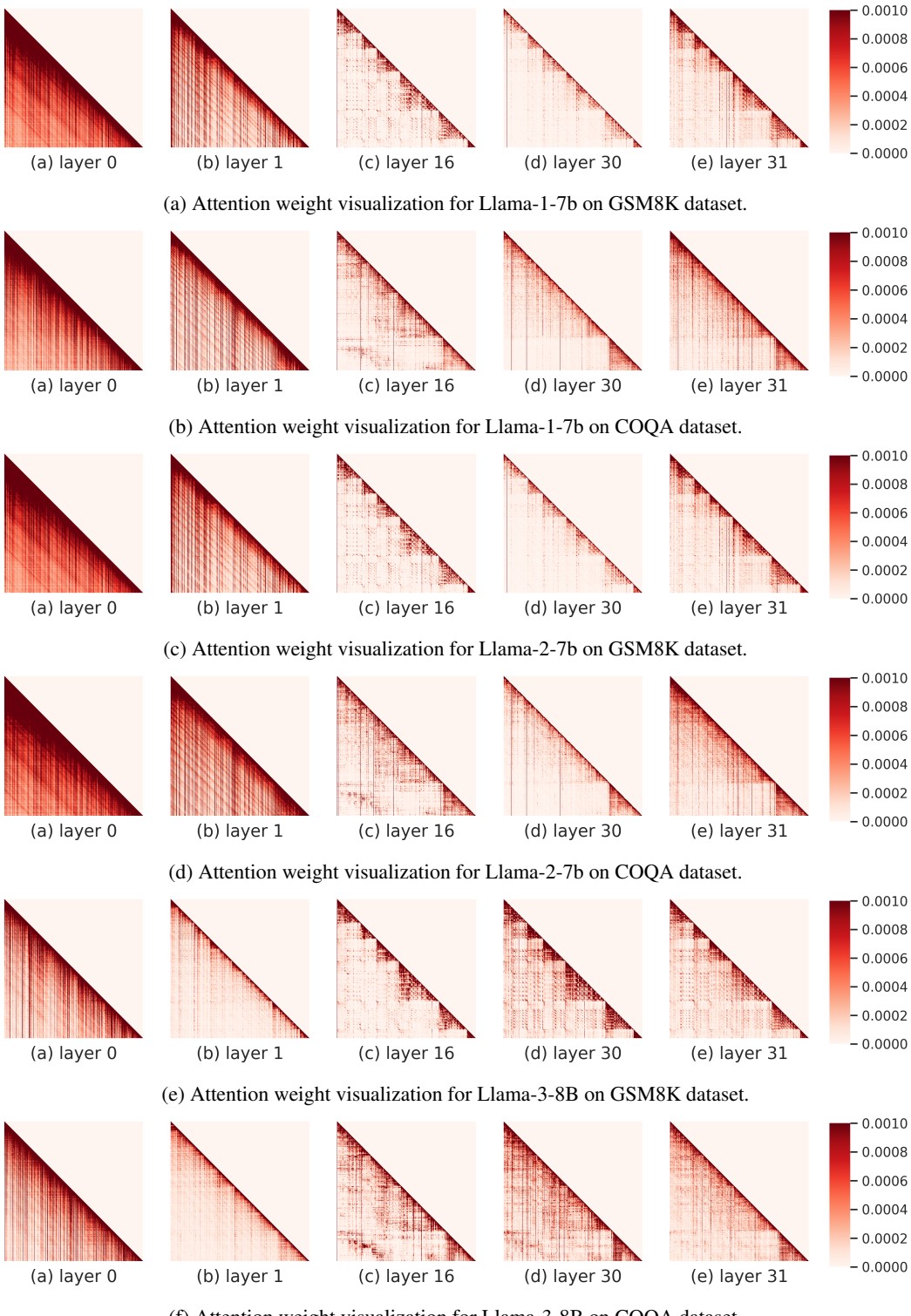

Figure 10: Attention weight visualization across various models and datasets.

where $S_l^h$ denotes the cache size allocated to the $h$-th attention head in the $l$-th layer, $S$ is the original cache size per head, and $\rho$ is the compression ratio. Our goal is to maximize the model's output

quality during inference by optimally allocating the limited cache resources among different layers and attention heads.

### A.9.2 ATTENTION VARIANCE AND INFORMATION ENTROPY

**Attention Weights as Probability Distributions.** For each attention head $h$ in layer $l$, the attention weight matrix $\mathbf{A}_p^{l,h} \in \mathbb{R}^{L_{\text{prompt}} \times L_{\text{prompt}}}$ is computed as:

$$\mathbf{A}_p^{l,h} = \text{Softmax}\left(\frac{\mathbf{Q}_p^{l,h}\mathbf{K}_p^{l,h\top}}{\sqrt{D}}\right), \tag{18}$$

where $\mathbf{Q}p^{l,h}$ and $\mathbf{K}_p^{l,h}$ are the query and key matrices for the prompt encoding, respectively, and $D$ is the dimensionality. The attention weights can be viewed as probability distributions over tokens:

$$p_i^{l,h} = \frac{\exp(e_i^{l,h})}{\sum_j \exp(e_j^{l,h})}, \tag{19}$$

where $e_i^{l,h}$ is the attention score for token $i$.

**Calculating Attention Variance and Information Entropy.** The attention variance for head $h$ in layer $l$ is:

$$\mathbf{F}_v^{l,h} = \text{Var}\left(\mathbf{s}^{l,h}\right), \text{ and } \mathbf{s}^{l,h} = \sum_{i=0}^{L_{\text{prompt}}} \mathbf{A}_p^{l,h}[i,:] \tag{20}$$

where $\mathbf{s}^{l,h}$ is the cumulative attention sequence obtained by summing the columns of $\mathbf{A}_p^{l,h}$. The information entropy for the attention distribution is:

$$H^{l,h} = -\sum_i p_i^{l,h} \log p_i^{l,h}. \tag{21}$$

Typically, a lower attention variance $\mathbf{F}_v^{l,h}$ indicates a more uniform distribution (higher entropy $H^{l,h}$), implying that information is more evenly distributed across tokens.

### A.9.3 INFORMATION RETENTION AND CACHE SIZE

The information retained by attention head $h$ in layer $l$ depends on its cache size $S_l^h$ and information entropy $H^{l,h}$:

$$I^{l,h} = f(S_l^h, H^{l,h}), \tag{22}$$

where $f$ is a monotonically increasing function, indicating that a larger cache size leads to higher information retention. The marginal information gain for head $h$ in layer $l$ is:

$$\Delta I^{l,h} = \frac{\partial I^{l,h}}{\partial S_l^h}. \tag{23}$$

During the inference phase, the attention weight distributions (and thus the information entropy $H^{l,h}$) are determined by the model's parameters and input data, not by the cache size $S_l^h$. Therefore, $H^{l,h}$ is treated as a constant and does not appear in $\Delta I^{l,h}$ because it does not depend on $S_l^h$.

### A.9.4 OPTIMIZATION OBJECTIVE

Our objective is to maximize the total information retention $Q$ under the cache size constraint:

$$\max_{S_l^h} \quad Q = \sum_{l=1}^{L}\sum_{h=1}^{H} I^{l,h} = \sum_{l=1}^{L}\sum_{h=1}^{H} f(S_l^h, H^{l,h}), \text{ s.t.} \qquad \sum_{l=1}^{L}\sum_{h=1}^{H} S_l^h = S_{\text{total}}. \tag{24}$$

To maximize $Q$ under the constraint, we use the Lagrange multiplier method from optimization theory. By introducing the Lagrange multiplier $\mu$, we construct the Lagrangian function:

$$\mathcal{L} = \sum_{l=1}^{L}\sum_{h=1}^{H} I^{l,h} - \mu\left(\sum_{l=1}^{L}\sum_{h=1}^{H} S_l^h - S_{\text{total}}\right). \tag{25}$$

To find the optimal cache allocation that maximizes $Q$, under the constraint, we take the partial derivative of the Lagrangian $L$ with respect to each $S_l^h$ and set it to zero:

$$\frac{\partial \mathcal{L}}{\partial S_l^h} = \frac{\partial I^{l,h}}{\partial S_l^h} - \mu = 0. \tag{26}$$

This yields the condition:

$$\frac{\partial I^{l,h}}{\partial S_l^h} = \mu, \quad \forall l, h. \tag{27}$$

This equation indicates that, at the optimal allocation, the marginal information gain for each attention head equals the Lagrange multiplier $\mu$. In other words, we achieve the maximum total information retention $Q$ when the marginal information gains are balanced across all attention heads.

**Optimal Allocation of Cache Size.** Under the constraint of limited total resources, if the marginal information gain of a particular attention head $\Delta I^{l,h}$ is greater than $\mu$, increasing its cache size will enhance the total information retention $Q$. Conversely, if the marginal information gain of an attention head $\Delta I^{l,h}$ is less than $\mu$, reducing its cache size and reallocating those resources to attention heads with higher marginal information gain will increase $Q$.

### A.9.5 DYNAMIC ALLOCATION STRATEGY.

**Cache Size Allocation Based on Attention Variance.** We propose allocating cache sizes inversely proportional to the attention variances:

$$S_l^h = \alpha_l^h \cdot S_{\text{total}}, \quad \text{where} \quad \alpha_l^h = \frac{\exp(-\mathbf{F}_v^{l,h})}{\sum_{l=1}^{L} \sum_{h=1}^{H} \exp(-\mathbf{F}_v^{l,h})}. \tag{28}$$

This strategy ensures that attention heads with lower variance (higher entropy) receive more cache, adhering to the principle of allocating more resources to where the marginal information gain is higher. The total cache size for layer $l$ with the layer-level allocation proportion are:

$$S_l = \sum_{h=1}^{H} S_l^h, \text{ and } \alpha_l = \sum_{h=1}^{H} \alpha_l^h. \tag{29}$$

**Limitations of Uniform Compression.** Under a uniform compression ratio $\rho$, each attention head receives the same cache size:

$$S_l^h = \rho \cdot S. \tag{30}$$

However, due to varying attention variances and entropies across different heads and layers, the marginal information gains $\Delta I^{l,h}$ are unequal. This imbalance leads to suboptimal total information retention compared to the dynamic allocation strategy.

### A.9.6 THEORETICAL PROOF OF SUPERIORITY

**Total Information Retention Comparison.**

*Dynamic Allocation Strategy.* The total information retention is:

$$Q_{\text{dynamic}} = \sum_{l=1}^{L} \sum_{h=1}^{H} I_{\text{dynamic}}^{l,h} = \sum_{l=1}^{L} \sum_{h=1}^{H} f\left(S_l^{h,\text{dynamic}}, H^{l,h}\right). \tag{31}$$

*Uniform Compression Strategy.* The total information retention is:

$$Q_{\text{uniform}} = \sum_{l=1}^{L} \sum_{h=1}^{H} I_{\text{uniform}}^{l,h} = \sum_{l=1}^{L} \sum_{h=1}^{H} f\left(\rho \cdot S, H^{l,h}\right). \tag{32}$$

Since the dynamic allocation strategy balances the marginal information gains across all attention heads, it maximizes the total information retention $Q_{\text{dynamic}}$. In contrast, the uniform compression

strategy cannot balance the marginal gains due to the diversity in $H^{l,h}$, resulting in $Q_{\text{uniform}} < Q_{\text{dynamic}}$. Next, we give a numerical example.

**Numerical Example.** Since $I^{l,h}$ is monotonically increasing function, we assume the information retention function of cache memory (Smith, 1982; Thomas & Joy, 2006) is:

$$I^{l,h} = H^{l,h} \left(1 - \exp\left(-kS_l^h\right)\right), \tag{33}$$

where $k$ is a positive constant. The marginal information gain is:

$$\Delta I^{l,h} = \frac{\partial I^{l,h}}{\partial S_l^h} = H^{l,h} k \exp\left(-kS_l^h\right). \tag{34}$$

Set the marginal gains equal:

$$H^{1,1} k \exp\left(-kS_1^1\right) = \cdots = H^{L,H} k \exp\left(-kS_L^H\right) = \mu. \tag{35}$$

Solving for $S_l^h$ yields:

$$S_l^h = -\frac{1}{k} \ln\left(\frac{\mu}{H^{l,h}k}\right). \tag{36}$$

The total cache size constraint is:

$$\sum_{l=1}^{L} \sum_{h=1}^{H} S_l^h = S_{\text{total}}. \tag{37}$$

By iteratively solving for $\mu$, we obtain the optimal cache sizes $S_l^h$.

**Comparison of Total Information Retention.**

*Dynamic Allocation.* The total information retention is:

$$Q_{\text{dynamic}} = \sum_{l=1}^{L} \sum_{h=1}^{H} H^{l,h} \left(1 - \exp\left(-kS_l^{h,\text{dynamic}}\right)\right). \tag{38}$$

*Uniform Compression.* The total information retention is:

$$Q_{\text{uniform}} = \sum_{l=1}^{L} \sum_{h=1}^{H} H^{l,h} \left(1 - \exp\left(-k\rho S\right)\right). \tag{39}$$

Since $S_l^{h,\text{dynamic}}$ is adjusted based on $H^{l,h}$, it follows that $Q_{\text{dynamic}} > Q_{\text{uniform}}$. By considering the attention variances and information entropies of individual attention heads, the dynamic KV cache allocation strategy effectively balances the marginal information gains, leading to maximal total information retention during inference. This theoretical analysis demonstrates that the dynamic allocation strategy outperforms the uniform compression strategy, as it optimally utilizes limited cache resources to enhance model output quality.

A.10 COMPARISON OF D$_2$O WITH OTHER NEW BASELINES

Table 18: Comparison of D$_2$O with other new methods in LongBench (20% KV cache).

| Model | | Qasper | QMSum | MultiNews | TREC | TriviaQA | SAMSum | Lcc | RB-P |
|---|---|---|---|---|---|---|---|---|---|
| | Full Cache | 29.8 | 24.44 | 26.28 | 66.67 | 86.16 | 41.11 | 56.91 | 49.09 |
| | SnapKV | 23.32 | 20.18 | 23.97 | 60.62 | 82.46 | 36.99 | 52.68 | 45.53 |
| | PyramidKV | 23.80 | 20.60 | 24.22 | 59.44 | 82.90 | 35.31 | 53.52 | 42.11 |
| Mistral-7B-Instruct-v0.2 | PyramidInfer | 23.38 | 19.80 | 21.23 | 58.35 | 78.24 | 35.18 | 52.35 | 41.88 |
| | SirLLMs | 21.48 | 18.81 | 23.12 | 54.79 | 78.57 | 31.72 | 53.42 | 43.54 |
| | LoCoco | 22.55 | 19.42 | **24.80** | 58.71 | 75.32 | 35.85 | 54.15 | 41.58 |
| | **D$_2$O** | **25.72** | **21.90** | 24.06 | **62.99** | **84.02** | **38.03** | **55.17** | **46.15** |

We also conducted additional comparisons with more contemporary baselines: token eviction approaches such as SnapKV (Li et al., 2024) and SirLLM (Yao et al., 2024), as well as token compression

techniques like LoCoCo (Cai et al., 2024a), and layer-wise KV cache compression methods like PyramidKV (Cai et al., 2024b) and PyramidInfer (Yang et al., 2024a). These experiments were performed on the LongBench dataset using the Mistral-7B model and followed the setting from Liu et al. (2024).

As shown in the Table 18, $D_2O$ consistently outperforms these baseline methods across most datasets. This demonstrates that $D_2O$'s combination of dynamic token-level merging and dynamic layer-wise cache allocation effectively preserves context and delivers robust performance in long-context scenarios.

### A.11 $D_2O$ Performance in 100K+ Long Contexts

To validate the capability of D2O in handling ultra-long contexts, we also conducted additional evaluations on datasets with longer contexts, including InfiniteBench (Zhang et al., 2024a) and RULER (Hsieh et al., 2024), both featuring instances exceeding 100K tokens. To ensure a fair comparison, we utilized the Llama-3.1-8B-Instruct (128K) model and benchmarked $D_2O$ against representative baselines, such as $H_2O$ (based on eviction), PyramidKV (layer-wise reduction of KV cache) and CaM (mergency of value tokens). All experiments were performed on four A100 GPUs with 80GB memory.

The results, provided in the tables 19 and 20, demonstrate that D2O consistently outperforms these baselines across both InfiniteBench and RULER. These findings highlight D2O's ability to effectively preserve context and maintain strong performance in extreme long-context scenarios, further validating its robustness.

Table 19: Comparison of $D_2O$ with other baselines in InfiniteBench (20% KV cache).

| Model | | R.PK | R.Num | R.KV | Choice | QA | Math.F | Code.Debug |
|---|---|---|---|---|---|---|---|---|
| | Full Cache | 100 | 99.52 | 28.66 | 62.15 | 25.78 | 35.86 | 26.94 |
| **Llama-3.1-8B-Instruct (128K)** | $H_2O$ | 62.15 | 63.84 | 18.85 | 51.42 | 16.13 | 27.65 | 19.35 |
| | PyramidKV | 65.25 | 73.88 | 19.82 | 52.91 | 19.36 | 28.19 | 24.08 |
| | CaM | 76.95 | 78.55 | 23.99 | 51.03 | 20.72 | 32.07 | 23.59 |
| | $D_2O$ | **86.75** | **91.35** | 25.88 | **54.35** | **24.84** | **33.85** | **25.54** |

Table 20: Comparison of $D_2O$ with other baselines in Ruler (20% KV cache).

| Model | | 4K | 8K | 16K | 32K | 64K | 128K |
|---|---|---|---|---|---|---|---|
| | Full Cache | 95.5 | 93.8 | 91.6 | 87.4 | 84.7 | 77.0 |
| **Llama-3.1-8B-Instruct (128K)** | $H_2O$ | 90.5 | 88.2 | 85.7 | 83.2 | 77.8 | 72.5 |
| | PyramidKV | 89.5 | 88 | 86.2 | 82.1 | 79.1 | 71.7 |
| | CaM | 91.6 | 86.6 | 85.4 | 81.8 | 79.6 | 72.6 |
| | $D_2O$ | **92.1** | **89.7** | **86.2** | **83.7** | **80.6** | **74.7** |

### A.12 Comparison of $D_2O$ Using Other LongBench Setting

To validate the effectiveness of $D_2O$ under other LongBench settings (Cai et al., 2024b), we conducted additional experiments, including using LLaMA-3-8B-Instruct and Mistral-7B-Instruct models as backbones, and conducted experiments under two KV cache scenarios (KV size = 64 and 2048). For D2O, the total compressed KV cache size was configured as 64×L or 2048×L, where L is the number of layers. For comparison, results of SnapKV Li et al. (2024), StreamingLLM Xiao et al. (2023a), $H_2O$ Zhang et al. (2024c), and PyramidKV Zhang et al. (2024c) were taken directly from the PyramidKV Cai et al. (2024b) paper. The evaluation for $D_2O$ was conducted using the PyramidKV-provided LongBench settings. The results demonstrate that under extreme compression settings (KV size = 64), $D_2O$ outperforms all baselines, highlighting the effectiveness of $D_2O$'s combination of layer-level dynamic KV cache allocation and token-level dynamic merging. With higher KV sizes (KV size = 2048), $D_2O$ still achieves better performance than baselines on most datasets, confirming the importance of our two-level operation approach.

Table 21: Performance evaluation of D$_2$O on various models in LongBench benchmarks following other setting (Cai et al., 2024b).

| Methods | Single-Document QA | | | Multi-Document QA | | | Summarization | | | Summarization | | | Synthetic | | Code | |
|---|---|---|---|---|---|---|---|---|---|---|---|---|---|---|---|---|
| | NrtvQA | Qasper | MF-en | HotpotQA | 2WikiMQA | Musique | GovReport | QMSum | MultiNews | TREC | TriviaQA | SAMSum | PCount | PRe | Lcc | RB-P |
| Full Model | 25.7 | 29.75 | 41.12 | 45.55 | 35.87 | 22.35 | 25.63 | 23.03 | 26.21 | 73.00 | 90.56 | 41.88 | 4.67 | 69.25 | 58.05 | 50.77 |
| **LLaMa-3-8B-Instruct (KV size = 64)** | | | | | | | | | | | | | | | | |
| SnapKV | 19.86 | 9.09 | 27.89 | 37.34 | 28.35 | **18.17** | 15.86 | 20.08 | 16.41 | 38.50 | 85.92 | 36.32 | 5.22 | 69.00 | 51.78 | 48.38 |
| StreamLLM | 17.44 | 8.68 | 22.25 | 35.37 | 31.51 | 15.97 | 15.46 | 20.06 | 14.64 | 38.00 | 72.33 | 29.10 | 5.42 | **69.50** | 46.14 | 45.09 |
| H2O | 20.80 | 11.34 | 27.03 | 37.25 | 30.01 | 17.94 | **18.29** | 21.49 | 19.43 | 38.40 | 84.70 | 37.76 | 5.65 | 69.33 | 53.44 | 50.15 |
| PyramidKV | 21.13 | 14.18 | 30.26 | 35.12 | 23.76 | 16.17 | 18.33 | 21.65 | 19.23 | 58.00 | 88.31 | 37.07 | 5.23 | **69.50** | 52.61 | 45.74 |
| D2O | **22.50** | **15.30** | **33.45** | **39.18** | **30.12** | 17.80 | **21.15** | **23.10** | **22.15** | 60.12 | **89.50** | **39.00** | 5.80 | 68.88 | **54.50** | **47.00** |
| **LLaMa-3-8B-Instruct (KV size = 2048)** | | | | | | | | | | | | | | | | |
| SnapKV | 25.86 | 29.55 | 41.10 | 44.99 | 35.80 | 21.81 | 25.98 | 23.40 | 26.46 | **73.50** | 90.56 | 41.66 | 5.17 | 69.25 | 56.65 | 49.94 |
| StreamLLM | 21.71 | 25.78 | 38.13 | 40.12 | 32.01 | 16.86 | 23.14 | 22.64 | 26.48 | 70.00 | 83.22 | 31.75 | 5.74 | 68.50 | 49.94 | 45.58 |
| H2O | 25.56 | 26.85 | 39.54 | 44.30 | 32.92 | 21.09 | 24.08 | 23.14 | 26.16 | 53.00 | 90.56 | 41.84 | 4.91 | 69.25 | 56.4 | 49.68 |
| PyramidKV | 25.40 | 29.71 | 40.25 | 44.76 | 35.32 | 21.98 | 23.30 | 23.30 | 26.19 | 73.00 | 90.56 | 42.14 | 5.22 | 69.25 | 58.76 | 51.18 |
| D2O | **25.92** | **30.01** | **41.25** | **45.32** | **35.97** | **22.45** | 25.78 | 23.21 | **26.55** | 73.20 | **90.60** | **42.02** | 5.35 | **69.54** | **58.8** | **51.22** |
| Full Model | 26.90 | 33.07 | 49.20 | 43.02 | 27.33 | 18.78 | 32.91 | 24.21 | 26.99 | 71.00 | 86.23 | 42.65 | 2.75 | 86.98 | 56.96 | 54.52 |
| **Mistral-7B-Instruct (KV size = 64)** | | | | | | | | | | | | | | | | |
| SnapKV | 16.94 | 17.17 | 39.51 | 36.87 | 22.26 | 15.18 | 14.75 | 20.35 | 21.45 | 37.50 | **84.16** | 37.28 | 4.50 | 61.13 | 42.40 | 38.44 |
| StreamLLM | 15.01 | 13.84 | 28.74 | 30.97 | 24.50 | 13.42 | 13.25 | 19.46 | 19.17 | 35.50 | 76.91 | 29.61 | **4.67** | 27.33 | 38.71 | 35.29 |
| H2O | 18.19 | 19.04 | 37.40 | 30.18 | 22.22 | 13.77 | 16.60 | 21.52 | 21.98 | 37.00 | 81.02 | **38.62** | 5.00 | 66.03 | 43.54 | 40.46 |
| PyramidKV | 20.91 | 20.21 | 39.94 | 33.57 | 22.87 | 15.70 | 17.31 | 21.23 | 21.41 | 54.00 | 81.98 | 36.96 | 3.58 | 60.83 | 44.52 | 37.99 |
| D2O | **22.50** | **22.38** | **41.26** | **38.15** | **23.50** | **16.20** | **23.54** | **22.50** | **22.00** | 55.00 | 82.50 | 38.00 | 2.32 | 67.00 | **47.53** | **41.81** |
| **Mistral-7B-Instruct (KV size = 2048)** | | | | | | | | | | | | | | | | |
| SnapKV | 25.89 | 32.93 | 48.56 | 42.96 | 27.42 | 19.02 | 26.56 | 24.47 | 26.69 | 70.00 | 86.27 | 42.57 | **5.50** | **88.90** | 50.42 | 46.72 |
| StreamLLM | 20.31 | 26.64 | 45.72 | 35.25 | 24.31 | 12.20 | 27.47 | 21.57 | 24.51 | 68.50 | 71.95 | 31.19 | 5.00 | 22.56 | 43.38 | 37.08 |
| H2O | 25.76 | 31.10 | **49.03** | 40.76 | 26.52 | 17.07 | 23.64 | 23.64 | 26.60 | 55.00 | **86.35** | 42.48 | **5.50** | 88.15 | 49.93 | 46.57 |
| PyramidKV | 25.53 | 32.21 | 48.97 | 42.26 | **27.50** | 19.36 | 23.93 | 23.97 | 26.73 | 71.00 | 86.25 | 42.94 | 4.50 | 87.90 | 53.12 | 47.21 |
| D2O | **26.18** | **33.12** | 48.88 | 43.50 | 27.38 | **19.45** | **33.20** | **24.55** | **26.88** | 71.50 | 86.12 | **43.10** | 5.35 | 87.21 | **57.10** | **54.75** |

