# OpenReview forum: "$\text{D}_{2}\text{O}$: Dynamic Discriminative Operations for Efficient Long-Context Inference of Large Language Models"
_ICLR.cc/2025/Conference — ICLR 2025 Poster_

### Official Review · Reviewer_aSVX · 2024-10-21

**Soundness:** 4
**Presentation:** 3
**Contribution:** 2
**Rating:** 6
**Confidence:** 4

**Summary:**

This paper introduces Dynamic Discriminative Operations (D₂O), a method to optimize KV cache management in large language models. At the layer level, D₂O allocates cache sizes dynamically based on attention variance. At the token level, it merges similar tokens and prunes less important ones, preserving essential contextual information. This approach reduces memory usage while improving efficiency and accuracy in long-text generation and inference tasks, outperforming existing cache compression techniques.

**Strengths:**

1. The proposed method outperforms traditional eviction-based approaches, significantly reducing performance loss in sparse long-text inference.
2. Extensive experimental analysis on hyperparameter selection, various models, and datasets strongly validates the method's effectiveness.
3. The method demonstrates excellent generalizability, as it can be applied to a wide range of existed pretrained models without requiring additional post-training or fine-tuning.

**Weaknesses:**

The innovation and novelty of the proposed method are somewhat limited compared to existing sparse attention pre-trained models or eviction-based inference approaches. For instance, the concept of token merging has been discussed in several prior works, such as https://openreview.net/pdf?id=LCTmppB165 and https://arxiv.org/html/2407.08454v1. Additionally, the idea of dynamically allocating different computational loads to different layers has also been explored in other works, such as https://arxiv.org/pdf/2404.02258.

**Questions:**

1. For the Dynamic Allocation Strategy, could it result in a single layer's computational load exceeding that of processing all tokens in a layer? How would this be handled in such cases?
2. In the Dynamic Allocation Strategy, how do you ensure that the computational load is uniform across samples within a batch? Could there be scenarios where some samples in the same batch have significantly higher or lower computational loads, thus reducing inference efficiency? Can this strategy be integrated with existing inference acceleration methods like vLLM?
3. In the Dynamic Allocation Strategy, is there any physical rationale behind the decision to allocate fewer KV caches to layers with high attention score variance?
4. Could the method be compared to a smaller full-size model with similar inference time costs?
5. In Table 9 and Figure 5, D₂O performs better than the full model in some tasks. Could you explain and analyze the reasons for this outcome?
6. In Section A.4, could you provide results for the full model in the multi-turn dialogue examples for comparison?
What are the inference costs in Table 3 for identical batch sizes?
7. Missing citation: https://arxiv.org/pdf/2408.03675 . This paper also focuses on optimizing token eviction methods which introduces a random eviction strategy that retains more token information.

---

> ### Author Response · Authors · 2024-11-19
> **Response to Reviewer aSVX (Part 1)**
>
> We would like to thank you for taking the time to review our work. Your questions and comments are pretty professional and enlightening. We address the questions and clarify the issues accordingly as described below.
>
> > **W1 The peculiarity of our innovation and novelty**
>
> Thank you for your feedback. We would like to clarify that, unlike the mentioned works, our approach uniquely combines layer-level and token-level adaptive operations for KV cache compression. Extensive experiments demonstrate that D2O achieves both efficient and effective compression compared to SOTA baselines.
>
> a. **Compared to CaM** [1], which uses accumulated attention scores and a Bernoulli mask for token merging, D2O sorts tokens by attention scores, calculates similarity, and applies an EMA threshold for selective merging. Additionally, D2O merges key caches first and applies weights to value caches, while CaM focuses solely on value caches.
>
> b. **Unlike KVMerger** [2], which uses a Gaussian kernel to merge all states within each set, D2O selectively merges tokens based on an EMA similarity threshold, filtering out redundant tokens during both prefill and generation phases. Moreover, KVMerger is a concurrent work.
>
> c. **Compared to MoD** [3], which trains a router to allocate tokens to specific layers, D2O is training-free, plug-and-play, and focuses on accurate layer-wise KV cache allocation instead of token distribution across layers.
> We will include these related works and comparisons in the revised paper to provide additional clarity.
>
> [1] https://openreview.net/pdf?id=LCTmppB165.
>
> [2] https://arxiv.org/html/2407.08454v1.
>
> [3] https://arxiv.org/pdf/2404.02258.
>
> > **Q1 Explanation of dynamic allocation strategy**
>
> Thank you for your question. We would like to clarify that such a situation will not occur. As detailed in Equation (5) and Appendix A.2, the total number of compressed KV tokens is given by:$\rho \cdot L \cdot S$, where $S$ is the length of the input prompt, $\rho$ is the KV cache compression ratio, and $L$ is the total number of layers. These tokens are dynamically allocated across layers.
> If the allocation for a single layer exceeds $S$ , we will set the KV cache size for that layer at $S$ , ensuring that no layer's computational load exceeds the cost of processing all tokens. This mechanism ensures both efficiency and scalability in our dynamic allocation strategy.
>
> > **Q2 Explanation of computational load**
>
> Thank you for your question. To ensure batch processing feasibility and maintain uniform computational load, we pad all sequences in the batch to match the length of the longest sequence. For layer-wise KV cache size allocation, we set the KV cache size for each layer in the batch to the maximum allocation determined by the most demanding (the largest cache size allocation) sample in that layer. This ensures that all samples in the batch have consistent KV cache sizes, enabling efficient parallel processing.
>
> It is worth noting that most of the inference datasets used in this paper default to single-sample inference in their benchmark settings. Additionally, while our current implementation has not been adapted to the vLLM framework, we plan to integrate D2O’s KV cache compression strategy with vLLMs in future work.
>
> > **Q3 The physical rationale behind the dynamic allocation strategy**
>
> Thank you for your question. The rationale behind allocating fewer KV caches to layers with high attention score variance is supported by the visualizations in Figures 1 and 9, as well as the attention variance statistics in Figure 4. These analyses indicate significant differences in attention density across layers:
>
> - In lower layers, attention maps are denser, reflecting more uniformly distributed information. Therefore, allocating more KV cache size in these layers reduces the risk of irreversible information loss during compression.
>
> - In higher layers, attention maps are sparser, indicating that information is concentrated on specific tokens. Consequently, fewer KV caches can be allocated to these layers without sacrificing critical information.
>
> This dynamic allocation approach ensures efficient KV cache usage while preserving essential context across different layers.

---

> ### Author Response · Authors · 2024-11-19
> **esponse to Reviewer aSVX (Part 2)**
>
> > **Q4 comparison with the smaller full-size model**
>
> Thank you for your suggestion. Due to differences in embedding sizes across LLMs of varying scales, the memory required for KV cache storage at the same sequence length differs significantly. This makes it challenging to select a smaller-scale LLM with similar inference time costs and conduct a fair comparison under KV cache compression scenarios.
>
> In most KV cache optimization studies, comparisons are typically made within the same model scale, applying different KV cache compression algorithms at a fixed compression ratio to ensure fairness in performance and throughput evaluation. The experimental setup in our paper follows the standard comparison methodology [1] [2], which ensures consistency and reliability in our results.
>
> [1] Heavy-Hitter Oracle for Efficient Generative Inference of Large Language Models.
>
> [2] Model Tells You What to Discard: Adaptive KV Cache Compression for LLMs.
>
> >**Q5 More analysis on Table 9 and Figure 5**
>
> Thank you for your question. Yes, this outcome is reasonable. For example, as shown in Figure 5 for the Llama-3-8B model, our dynamic token-level strategy selectively merges evicted tokens based on a similarity threshold. This process removes redundant information while preserving critical context, enabling the model to focus more effectively on the retained key tokens. As a result, the model can perform more accurate reasoning, leading to better performance compared to the full cache and other baselines under certain KV cache compression rates.
>
> Similar phenomena have also been reported in prior works, such as H2O [1], PyramidInfer [2], and FastGen [3], further validating this observation.
>
> [1] Heavy-Hitter Oracle for Efficient Generative Inference of Large Language Models.
>
> [2] PyramidInfer: Pyramid KV Cache Compression for High-throughput LLM Inference.
>
> [3] Model Tells You What to Discard: Adaptive KV Cache Compression for LLMs.
>
> > **Q6 More experimental results**
>
> Thank you for your comment. We followed the StreamLLMs codebase for benchmarking sample generation. In a single A100 GPU setup, the full model often encounters Out of Memory (OOM) issues during ultra-long text inference without KV cache optimization. In the updated paper, we will include the full model's output in the Appendix. If OOM occurs for a sample, we will indicate it accordingly.
>
> For Table 3, using a four-A100 80GB setup, our inference speed falls between SnapKV and CaM, while achieving the best performance among the methods. Thank you for highlighting this.
>
> | Methods        | L=50k (KV cache size = 4096)    | L=100k (KV cache size = 4096)   | L=50k (KV cache size = 8192)    | L=100k (KV cache size = 8192)   |
> |----------------|---------------------------------|---------------------------------|---------------------------------|---------------------------------|
> | Full Model     | 97.88 (93.6 min)               | 94.46 (175.4 min)              | 97.88 (93.6 min)               | 94.46 (175.4 min)              |
> | StreamingLLM   | 58.64 / 53.49 min              | 47.93 / 100.23 min             | 62.84 / 64.17 min              | 51.34 / 110.92 min             |
> | H2O            | 79.84 / 69.33 min              | 69.81 / 125.93 min             | 82.32 / 75.17 min              | 72.34 / 138.88 min             |
> | SnapKV         | 83.55 / 78.00 min              | 76.22 / 146.17 min             | 86.63 / 86.82 min              | 80.42 / 154.34 min             |
> | CaM            | 82.66 / 66.73 min              | 78.22 / 116.24 min             | 87.59 / 75.12 min              | 78.88 / 129.80 min             |
> | D2O            | 91.27 / 71.53 min              | 87.74 / 129.85 min             | 94.48 / 79.37 min              | 91.88 / 142.22 min             |
>
> > **Q7 Additional citation**
>
> Thank you for pointing this out. We will include a citation to this concurrent work in the related work section of the updated paper.
>
> You have offered constructive and valuable suggestions, making our submission more solid and complete. Once again, we sincerely express our best gratitude for your effort and time!

---

> ### Author Response · Authors · 2024-11-25
> **Thank you, Reviewer aSVX**
>
> Dear Reviewer aSVX,
>
> Thank you for your insightful comments and valuable contributions to our work. Your feedback and support have significantly enhanced the quality of our paper. Wishing you all the best in your endeavors!
>
> Bests,
>
> Your friends,
>
> Team 5363

---

### Official Review · Reviewer_FB7s · 2024-11-02

**Soundness:** 2
**Presentation:** 3
**Contribution:** 3
**Rating:** 6
**Confidence:** 5

**Summary:**

This paper explores methods for saving the KV cache during inference of large language models. Specifically, the authors investigate ignoring less important tokens in the context during decoding and find that, to prevent excessive performance degradation, the number of tokens discarded should vary across different layers. Additionally, they explore a token compensation mechanism to mitigate the information loss caused by discarding tokens. Experiments demonstrate that their approach accelerates the model while maintaining performance.

**Strengths:**

* The motivation is clear and saving the KV cache of LLMs during inference is significant.
* This paper deeply investigates the difference in token eviction across layers and explores a reasonable token compensation mechanism.
* The experiments are relatively extensive.

**Weaknesses:**

* Some judgments are inaccurate. As noted in lines 181-182: "Employing a uniform eviction strategy such as the ones proposed in H2O...". However, to my knowledge, H2O also adopts a layer-wise token eviction strategy, as evidenced in its open-source code [1].
* The baselines are limited. Exploring the issue of token eviction ratios across different layers is highly meaningful. However, to my knowledge, some relatively classic works in this area, such as PyramidKV [2], were not considered by the authors.
* Some experimental results are quite unusual. In detail,
  * Logically, discarding some tokens would lead to information loss. However, some results in Figure 5 of the paper indicate that, after discarding certain tokens, the performance improves significantly compared to the original model (for instance, on the TruthfulQA task). Is this reasonable?
  * The evaluation results on Longbench differ significantly from those in some previous studies. For example, The results in line 423 of the paper differ by more than **10** points from those in Table 1 of PyramidKV [2] and SnapKV [3]. I believe the authors need to carefully review and revise their findings to provide more reliable results.

[1] https://github.com/FMInference/H2O/blob/main/h2o_hf/utils_real_drop/modify_llama.py#L357

[2] Zhang, Yichi, et al. "PyramidKV: Dynamic KV Cache Compression based on Pyramidal Information Funneling." arXiv preprint arXiv:2406.02069 (2024).

[3] Li, Yuhong, et al. "Snapkv: Llm knows what you are looking for before generation." arXiv preprint arXiv:2404.14469 (2024).

**Questions:**

* In formula (5), $S_\ell$ is the cache size for each layer. However, when calculating $\alpha_\ell$, the number of model layers $L$ is multiplied again. Is this a typographical error?
* In formula (6), I cannot understand the calculation of AttnScore during token generation.

---

> ### Author Response · Authors · 2024-11-19
> **Response to Reviewer FB7s (Part 1)**
>
> We would like to express our most sincere gratitude for your time and effort. We value each of your suggestions and provide the following responses:
>
> > **W1: Clarification of statement in our paper**
>
> Thank you for your feedback. We respectfully think that perhaps you have misunderstood us. We would like to clarify a potential misunderstanding regarding our statement in Lines 181-182. When we refer to a "uniform eviction strategy," we mean that methods like H2O and SnapKV allocate the same KV cache size across all layers, regardless of the input or layer-specific representation. In contrast, D2O introduces an adaptive dynamic cache allocation strategy that adjusts KV cache size based on attention variance across layers, allowing for a more efficient and data-driven distribution of memory resources. We will also consider refining our presentation to “uniform layer-wise size” to make the statement clearer and more accurate.
>
> > **W2: Additional comparison with other methods.**
>
> Thank you for your feedback. To address your suggestion, we added comparisons with PyramidKV [1] and PyramidInfer [2] using the Mistral-7B-Instruct-v0.2 model on the LongBench dataset as the following table. The results show that D2O consistently outperforms these non-adaptive KV cache allocation methods across most datasets, demonstrating the effectiveness of our inverse variance softmax-based dynamic allocation strategy. This approach excels in long-context scenarios by preserving critical information across layers.
>
> #### LongBench (Mistral-7B-Instruct-v0.2) / (20% KV Cache)
> |Model|Qasper|QMSum|MultiNews|TREC|TriviaQA|SAMSum|LCC|RepoBench-P|
> |-|-|-|-|-|-|-|-|-|
> |Full_cache|29.8|24.44|26.28| 66.67|86.16|41.11|56.91|49.09|
> |PyramidKV|23.8|20.6|**24.22**|59.44|82.9|35.31|53.52|42.11|
> |PyramidInfer|23.38|19.8|21.23|58.35|78.24|35.18|52.35 |41.88|
> |D2O|**25.72**|**21.9**|24.06|**62.99**|**84.02**|**38.03**|**55.17**|**46.15**|
>
> Additionally, we also want to clarify that D2O differs from the heuristic approach of PyramidKV or PyramidInfer, which reduces cache size uniformly with increasing layer depth. D2O adapts dynamically to the input data, leveraging attention variance to allocate KV cache size more precisely, leading to better performance.
>
> > **W3: More analysis on our experimental results**
>
> + Thank you for your question. Yes, this is a reasonable outcome. In the Llama-3-8B model scenario shown in Figure 5, our dynamic token-level strategy uses similarity thresholds to selectively merge evicted tokens. This process removes redundant information while preserving key context, enabling the model to focus more effectively on critical tokens and perform more accurate reasoning. As a result, under certain KV cache compression rates, D2O can achieve better performance than the full cache and other baselines.
> Similar phenomena have also been reported in prior works, such as PyramidInfer [2],  H2O [3], and FastGen [4], further validating this observation.
>
> + Thank you for your feedback. We would like to clarify that our evaluation of LongBench was conducted using the official code from KIVI's GitHub repository [5][6]. Our implementation results align closely with the full cache results of Llama-2-7B/13B/ Mistral-7B reported in the KIVI paper, as shown in the table below.
>
> #### Long-bench implementation result comparison
> | | Qasper | QMSum | MultiNews | TREC| TriviaQA | SAMSum | LCC | RepoBench-P |
> |-|-|-|-|-|-|-|-|-|
> |Llama-2-13b| | | | | | | | |
> | KIVI's report | 9.32 |   21.38 | 3.71  | 70| 87.87 | 43.55 | 66.61| 56.42 |
> | Ours | 9.37  |   21.37  | 4.74  | 63.33 | 87.37  | 42.3  | 67.36 | 54.62 |
> | Mistral-7B-Instruct-v0.2| | | | | | | | |
> | KIVI's report | 29.41  |   23.99  | 27.09 | 71  | 86.23  | 43.04 | 53.49| 51.4  |
> | Ours | 29.8  |  24.44 | 26.28 | 66.67 | 86.16  | 41.11 | 56.91 | 49.09 |
> | Llama-2-7b| | | | | | | | |
> | KIVI's report| 9.52 |  21.28 | 3.51  | 66 | 87.72  | 41.69 | 66.66  | 59.82 |
> | Ours | 8.92 |  21.29 | 1.42  | 61  | 89.81  | 39.73 | 67.95 | 55.14 |
>
> For our LongBench experiments, we ensured fairness by using the official code and default configurations provided for each baseline method. The differences in reported results may arise from variations in benchmark code versions or experimental setups between studies. For example, we use 20% KV cache size based on the input prompt, whereas they use a fixed context size (e.g., 2048). We will clarify this distinction in the revised paper to avoid confusion.
>
>
> [1] PyramidKV: Dynamic KV Cache Compression based on Pyramidal Information Funneling.
>
> [2] PyramidInfer: Pyramid KV Cache Compression for High-throughput LLM Inference.
>
> [3] Heavy-Hitter Oracle for Efficient Generative Inference of Large Language Models.
>
> [4] Model Tells You What to Discard: Adaptive KV Cache Compression for LLMs.
>
> [5] KIVI: A Tuning-Free Asymmetric 2bit Quantization for KV Cache. ICML 2024
>
> [6] https://github.com/jy-yuan/KIVI/blob/main/eval_long_bench.py.

---

> ### Author Response · Authors · 2024-11-19
> **Response to Reviewer FB7s (Part 2)**
>
> > **Q1: Clarification of the formula (5)**
>
> Thank you for your question. We would like to clarify that this is not a typographical error. The detailed derivation process is provided in Appendix A.2. In the formula, $\rho$ represents the fixed KV cache compression ratio per layer (e.g., 20%), and $L$ is the number of layers. Using the method outlined in the formula, we dynamically allocate the total KV cache budget across layers based on the inverse-softmax distribution. This ensures that each layer receives an appropriate cache size proportional to its attention variance.
>
> > **Q2: More explanation on calculating AttnScore during token generation**
>
> Thank you for your question. To clarify the calculation of $\text{AttnScore}$ in formula (6) during token generation:
> At step $i$ in the generation phase, the hidden state of the newly generated token has a size of $(\text{batch}, 1, \text{dim})$. After multiplying it with the query weight matrix $W_q$, we obtain $q_i$, an embedding of size $(\text{batch}, 1, \text{dim})$. This hidden state is then used to compute new $k_i$ and $v_i$ embeddings by multiplying it with $W_k$ and $W_v$, respectively. These embeddings are subsequently stored in the KV cache, and the KV cache is updated, extending the key and value token lengths to $L_k + 1$.
>
> Next, $q_i$ is multiplied with the Key embeddings in the KV cache (size: $\text{batch}, L_k + 1, \text{dim}$), resulting in an attention score of size $(\text{batch}, 1, L_k + 1)$. Since we compute cumulative attention scores, this new attention score is added to the scores obtained during the prompt encoding phase, yielding the updated attention score for the generation phase.
>
> We hope this explanation clarifies the calculation process. Please let us know if further details are needed.
>
> You have offered constructive and valuable suggestions, making our submission more solid and complete.We promise to include the above results and clearer expression in the final version. Once again, we sincerely express our best gratitude for your effort and time!
>
> Best wishes,
>
> All authors of Submission 5363.

---

> ### Author Response · Authors · 2024-11-23
> **Response to Reviewer FB7s (New Part)**
>
> Dear Reviewer FB7s:
>
> Thank you for your thoughtful follow-up comments. We greatly appreciate the opportunity to address your remaining concerns, as your feedback has been instrumental in improving our work. Below, we provide detailed responses to your points and outline the steps we have taken to address your suggestions:
>
> **1. Aligning Evaluation using Longbench code from PyramidKV :**
>
> Following your suggestion, we have evaluated our D2O method using the PyramidKV codebase for LongBench experiments. We strictly adhered to the settings provided in the official code, including using LLaMA-3-8B-Instruct and Mistral-7B-Instruct models as backbones, and conducted experiments under two KV cache scenarios (KV size = 64 and 2048). For D2O, the total compressed KV cache size was configured as 64×$L$ or 2048×$L$, where L is the number of layers.
>
> For comparison, results of SnapKV, StreamLLM, H2O, and PyramidKV were taken directly from the PyramidKV paper. The evaluation for D2O was conducted using the PyramidKV-provided LongBench settings. From the following Table, the results demonstrate that under extreme compression settings (KV size = 64), D2O outperforms all baselines in average, highlighting the effectiveness of D2O’s combination of layer-level dynamic KV cache allocation and token-level dynamic merging. With higher KV sizes (KV size = 2048), D2O still achieves better performance than baselines on most datasets, confirming the consistent performance of our two-level operation approach.
>
>
> ### LLaMa-3-8B-Instruct (KV size = 64)
> | Model      | NQ   | QP   | MF   | HQ   | 2WQ  | MQ   | GR   | QM   | MN   | TR   | TQ   | SS    | PC   | PR   | LC   | RP   | AVG   |
> |------------|------|------|------|------|------|------|------|------|------|------|------|-------|------|------|------|------|-------|
> | Full Cache | 25.7 | 29.75| 41.12| 45.55| 35.87| 22.35| 25.63| 23.03| 26.21| 73   | 90.56| 41.88 | 4.67 | 69.25| 58.05| 50.77| 41.46 |
> | SnapKV     | 19.86| 9.09 | 27.89| 37.34| 28.35| **18.17**| 15.86| 20.08| 16.41| 38.5 | 85.92| 36.32 | 5.22 | 69   | 51.78| 48.38| 33.05 |
> | StreamLLM  | 17.44| 8.68 | 22.25| 35.37| 31.51| 15.97| 15.46| 20.06| 14.64| 38   | 72.33| 29.1  | 5.42 | **69.5** | 46.14| 45.09| 30.43 |
> | H2O        | 20.8 | 11.34| 27.03| 37.25| 30.01| 17.94| 18.29| 21.49| 19.43| 38.4 | 84.7 | 37.76 | **5.65** | 69.33| 53.44| 50.15| 33.89 |
> | PyramidKV  | 21.13| 14.18| 30.26| 35.12| 23.76| 16.17| 18.33| 21.65| 19.23| 58   | 88.31| 37.07 | 5.23 | **69.5** | 52.61| 45.74| 34.76 |
> | D2O        | **22.5** | **15.3** | **33.45**| **39.18**| **30.12**| 17.8 | **21.15**| **23.1** | **22.15**| **60.12**| **89.5** | **39**    | **5.8**  | 68.88| **54.5** | **47**   | **36.85** |
>
>
> ### LLaMa-3-8B-Instruct (KV size = 2048)
> | Model      | NQ   | QP   | MF   | HQ   | 2WQ  | MQ   | GR   | QM   | MN   | TR   | TQ   | SS    | PC   | PR   | LC   | RP   | AVG   |
> |------------|------|------|------|------|------|------|------|------|------|------|------|-------|------|------|------|------|-------|
> | Full Cache | 25.7 | 29.75| 41.12| 45.55| 35.87| 22.35| 25.63| 23.03| 26.21| 73   | 90.56| 41.88 | 4.67 | 69.25| 58.05| 50.77| 41.46 |
> | SnapKV     | 25.86| 29.55| 41.1 | 44.99| 35.8 | 21.81| **25.98**| **23.4**| 26.46| **73.5** | 90.56| 41.66 | 5.17 | 69.25| 56.65| 49.94| 41.35 |
> | StreamLLM  | 21.71| 25.78| 38.13| 40.12| 32.01| 16.86| 23.14| 22.64| 26.48| 70   | 83.22| 31.75 | 5.74 | 68.5 | 53.5 | 45.58| 37.82 |
> | H2O        | 25.56| 26.85| 39.54| 44.3 | 32.92| 21.09| 24.08| 23.14| 26.16| 53   | 90.56| 41.84 | 4.91 | 69.25| 56.4 | 49.68| 39.35 |
> | PyramidKV  | 25.4 | 29.71| 40.25| 44.76| 35.32| 21.98| 23.3 | 23.3 | 26.19| 73   | 90.56| 42.14 | 5.22 | 69.25| 58.76| 51.18| 41.49 |
> | D2O        | **25.92**| **30.01**| **41.25**| **45.32**| **35.97**| **22.45**| 25.78| 23.21| **26.55**| 73.2 | **90.6** | **42.02**| **5.35** | **69.54**| **58.8**| **51.22**| **41.7** |

---

> ### Author Response · Authors · 2024-11-23
> **Response to Reviewer FB7s (New Part 2)**
>
> ### Mistral-7B-Instruct (KV size = 64)
>
> | Model      | NQ   | QP   | MF   | HQ   | 2WQ  | MQ   | GR   | QM   | MN   | TR   | TQ   | SS    | PC   | PR   | LC   | RP   | AVG   |
> |------------|------|------|------|------|------|------|------|------|------|------|------|-------|------|------|------|------|-------|
> | Full Cache | 26.9 | 33.07| 49.2 | 43.02| 27.33| 18.78| 32.91| 24.21| 26.99| 71   | 86.23| 42.65 | 2.75 | 86.98| 56.96| 54.52| 42.71 |
> | SnapKV     | 16.94| 17.17| 39.51| 36.87| 22.26| 15.18| 14.75| 20.35| 21.45| 37.5 | **84.16**| 37.28 | 4.5  | 61.13| 42.4 | 38.44| 30.72 |
> | StreamLLM  | 15.01| 13.84| 28.74| 30.97| 24.5 | 13.42| 13.25| 19.46| 19.17| 35.5 | 76.91| 29.61 | **4.67** | 27.33| 38.71| 35.29| 25.6  |
> | H2O        | 18.19| 19.04| 37.4 | 30.18| 22.22| 13.77| 16.6 | 21.52| 21.98| 37   | 81.02| 38.62| 5    | 66.03| 43.54| 40.46| 30.88 |
> | PyramidKV  | 20.91| 20.21| 39.94| 33.57| 22.87| 15.7 | 17.31| 21.23| 21.41| 54   | 81.98| 36.96 | 3.58 | 60.83| 44.52| 37.99| 32.19 |
> | D2O        | **22.5** | **22.38**| **41.26**| **38.15**| **23.5** | **16.2** | **23.54**| **22.5** | **22** | **55** | 82.5 | 38   | 2.32 | 67   | **47.53**| **41.81**| **35.38** |
>
> ### Mistral-7B-Instruct (KV size = 2048)
> | Model      | NQ   | QP   | MF   | HQ   | 2WQ  | MQ   | GR   | QM   | MN   | TR   | TQ   | SS    | PC   | PR   | LC   | RP   | AVG   |
> |------------|------|------|------|------|------|------|------|------|------|------|------|-------|------|------|------|------|-------|
> | Full Cache | 26.9 | 33.07| 49.2 | 43.02| 27.33| 18.78| 32.91| 24.21| 26.99| 71   | 86.23| 42.65 | 2.75 | 86.98| 56.96| 54.52| 42.71 |
> | SnapKV     | 25.89| 32.93| 48.56| 42.96| 27.42| 19.02| 26.56| 24.47| 26.69| 70   | 86.27| 42.57 | **5.5**  | **88.9** | 50.42| 46.72| 41.56 |
> | StreamLLM  | 20.31| 26.64| 45.72| 35.25| 24.31| 12.2| 27.47| 21.57| 24.51| 68.5 | 71.95| 31.19 | 5    | 22.56| 43.38| 37.08| 32.35 |
> | H2O        | 25.76| 31.1 | **49.03**| 40.76| 26.52| 17.07| 23.64| 23.64| 26.6 | 55   | **86.35**| 42.48 |**5.5**  | 88.15| 49.93| 46.57| 39.95 |
> | PyramidKV  | 25.53| 32.21| 48.97| 42.26| **27.5** | 19.36| 23.93| 23.97| 26.73| 71   | 86.12| 42.94 | 4.5  | 87.9 | 53.12| 47.21| 41.63 |
> | D2O        | **26.18**| **33.12**| 48.88| 43.5 | 27.38| **19.45**| **33.2**| **24.55**| **26.88**| **71.5** | 86.12| **43.1** | 5.35| 87.21| **57.1**| **54.75**| **43.02** |
>
> We observe that discrepancies may exist between KIVI [1] and PyramidKV LongBench setting [2] due to differences in data processing and experimental configurations. However, D2O achieves consistent performance across both settings. Furthermore, results on datasets requiring commonsense and complex reasoning (e.g., GSM8K, CoQA, TruthfulQA), as well as long-context datasets (Needle-in-a-Haystack, Ruler, InfiniteBench), validate the robustness of D2O’s two-level strategy. To ensure transparency, we have added the PyramidKV-based LongBench results to the appendix in the revised paper (Appendix A12).
>
> [1] https://github.com/jy-yuan/KIVI/blob/main/eval_long_bench.py
> [2] https://github.com/Zefan-Cai/KVCache-Factory/blob/main/run_longbench.py
>
> **2. For question 1**, regarding the absence of NarrativeQA and HotpotQA in the evaluation, we selected datasets aligned with KIVI’s Table 4 experimental settings, including Qasper, QMSum, MultiNews, TREC, TriviaQA, SAMSum, LCC, and RepoBench-P. While we did not include NarrativeQA and HotpotQA in this specific evaluation, Table 2 in the main paper already provides results for all datasets with multiple LLM backbones.
>
> **3. For question 2**, due to time and computational constraints during the rebuttal period, we used Mistral-7B-Instruct as the primary backbone for LongBench evaluations. However, we have also evaluated other backbones, such as LLaMA-3.1-8B-Instruct, on additional datasets like Ruler and InfiniteBench (Appendix A11). These results demonstrate that D2O’s effectiveness generalizes across different LLM backbones and datasets.
>
> We sincerely thank you for your suggestions, which have significantly strengthened our paper. We hope our responses and additional experiments adequately address your concerns, and we look forward to any further feedback you may have.
>
> Best regards,
>
> Your friends,
>
> Team 5363

---

> ### Author Response · Authors · 2024-11-23
> **Thank you, Reviewer FB7s**
>
> Dear Reviewer FB7s
>
> Thank you for your valuable feedback on our paper. We have incorporated your suggestions and experiments into the main text and the appendix. You are an outstanding reviewer, and under your guidance, our paper will improve significantly. We sincerely appreciate your efforts and wish you success in your future research. Merry Christmas!
>
> Best wishes，
>
> Your friend，
>
> Team 5363

---

### Official Review · Reviewer_VHpa · 2024-11-03

**Soundness:** 3
**Presentation:** 3
**Contribution:** 3
**Rating:** 6
**Confidence:** 3

**Summary:**

This paper focuses on token eviction for large language models and introduces a dynamic allocation strategy at the layer level and a compensation mechanism at the token level.

**Strengths:**

1. The proposed token eviction mechanisms at both the layer level and token level are well-reasoned and contribute to performance improvements in the models.
2. The effectiveness of the proposed methods is validated across multiple model backbones and long-context benchmarks.

**Weaknesses:**

1. The experimental comparisons are not comprehensive enough. The authors should compare their method with recent token eviction approaches, such as SnapKV and SirLLM. Additionally, comparisons with token compression techniques like LoCoco would be beneficial.
2. It is suggested that the authors evaluate the performance of their method on benchmarks with longer contexts, such as InfiniteBench and RULER, which consist of instances up to 128K tokens.

**Questions:**

Please refer to Weaknesses.

---

> ### Author Response · Authors · 2024-11-19
> **Response to Reviewer VHpa**
>
> We would like to thank you for taking the time to review our work. Your questions and comments are enlightening. We value each of them and have provided detailed responses below:
>
> > **W1: Additional comparison with recent token eviction approaches**
>
> Thank you for the valuable suggestion. Following your recommendation, we conducted additional comparisons with recent token eviction approaches such as SnapKV and SirLLM, as well as token compression techniques like LoCoco. These experiments were performed on the LongBench dataset using the Mistral-7B-Instruct-v0.2 model.
> As shown in the table below, D2O consistently outperforms these baseline methods across most datasets. This demonstrates that D2O’s combination of dynamic token-level merging and dynamic layer-wise cache allocation effectively preserves context and delivers robust performance in long-context scenarios.  We will incorporate the related works and comparative experiments you suggested in the updated version of our paper.
>
>
> #### LongBench (Mistral-7B-Instruct-v0.2) / (20% KV Cache)
> |             | Qasper | QMSum | MultiNews | TREC  | TriviaQA | SAMSum | LCC   | RepoBench-P |
> |-------------|--------|-------|-----------|-------|----------|--------|-------|-------------|
> | Full_cache  | 29.8   | 24.44 | 26.28     | 66.67 | 86.16    | 41.11  | 56.91 | 49.09       |
> | SnapKV      | 23.32  | 20.18 | 23.97     | 60.62 | 82.46   | 36.99  | 52.68 | 45.53        |
> | SirLLMs     | 21.48  | 18.81 | 23.12     | 54.79 | 78.57    | 31.72  | 53.42 | 43.54       |
> | LoCoco      | 22.55  | 19.42 | **24.8**      | 58.71 | 75.32    | 35.85  | 54.15 | 41.58       |
> | D2O         | **25.72**  | **21.9**  | 24.06     | **62.99** | **84.02**    | **38.03**  | **55.17** | **46.15**       |
>
> > **W2:Additional results on benchmarks with longer contexts**
>
> Thank you for your valuable suggestion. In response, we conducted additional evaluations on datasets with longer contexts, including InfiniteBench [1][2] and RULER [3], both featuring instances exceeding 100K tokens. To ensure a fair comparison, we utilized the Llama-3.1-8B-Instruct (128K) model and benchmarked D2O against representative baselines, such as H2O (eviction-based), PyramidKV (layer-wise KV cache reduction), and CaM (value token merging). All experiments were performed on four A100 GPUs with 80GB memory.
> The results, provided in the tables below, demonstrate that D2O consistently outperforms these baselines across both InfiniteBench and RULER. These findings highlight D2O’s ability to effectively preserve context and maintain strong performance in extreme long-context scenarios, further validating its robustness. We appreciate your feedback, which helped enhance our evaluation.
>
> [1] https://github.com/OpenBMB/InfiniteBench.
>
> [2] https://github.com/thunlp/InfLLM.
>
> [3] https://github.com/NVIDIA/RULER.
>
> #### Infinite-Bench (Llama-3.1-8B-Insct (128K)) / (20% KV Cache)
> |             | Passkey | Number_string | Kv_retrieval | Lookbook_choice_eng | Lookbook_qa_eng | Math.F  | code_debug |
> |-------------|---------|---------------|--------------|---------------------|------------------|---------|------------|
> | Full_Cache  | 100     | 99.52         | 28.66        | 62.15               | 25.78           | 35.86   | 26.94   |
> | H2O         | 62.15   | 63.84         | 18.85        | 51.42               | 16.13           | 27.65   | 19.35     |
> | PyramidKV   | 65.25   | 73.88         | 19.82        | 52.91               | 19.36           | 28.19   | 24.08  |
> | CaM         | 76.95   | 78.55         | 23.99        | 51.03               | 20.72           | 32.07   | 23.59      |
> | D2O         | **86.75**   | **91.35**         | **25.88**       | **54.35**               | **24.84**           | **33.85**   | **25.54**      |
>
>
> #### Ruler (Llama-3.1-8B-Instruct (128K)) / (20% KV Cache)
> |             | 4K  | 8K  | 16K | 32K  | 64K  | 128K |
> |-------------|-----|-----|-----|------|------|------|
> | Full_Cache  | 95.5| 93.8| 91.6| 87.4 | 84.7 | 77   |
> | H2O         | 90.5| 88.2| 85.7| 83.2 | 77.8 | 72.5 |
> | PyramidKV   | 89.5| 88  | 86.2| 82.1 | 79.1 | 71.7 |
> | CaM         | 91.6| 86.6| 85.4| 81.8 | 79.6 | 72.6 |
> | D2O         | **92.1**| **89.7**| **86.2**| **83.7** | **80.6** | **74.7** |
>
> You have offered constructive and valuable suggestions, making our submission more solid and complete. We promise to include the above results and analyses in the final version. Once again, we sincerely express our best gratitude for your effort and time!
>
> Best wishes,
>
> All authors of Submission 5363.

---

### Official Review · Reviewer_LdwY · 2024-11-03

**Soundness:** 2
**Presentation:** 2
**Contribution:** 2
**Rating:** 5
**Confidence:** 4

**Summary:**

This paper proposes D2O, a method for efficient LLM inference through dynamic KV cache optimization. The key contributions include: (1) layer-wise cache allocation based on attention density patterns, and (2) a token-level strategy combining eviction with dynamic merging using EMA threshold. The method achieves 3x throughput improvement while maintaining generation quality across various tasks.

**Strengths:**

-The paper brings up a major snag in putting LLMs into action, especially the memory requirements of the KV cache during making inferences. This is a crucial issue for real-world applications.

-The authors have conducted extensive experiments across multiple LLM architectures  and tasks, providing a broad evaluation of the method's effectiveness.

 -D2O shows notable improvements in throughput and memory efficiency, achieving up to 3x higher throughput while preserving generation quality, which are important metrics for the deployment of LLMs.

**Weaknesses:**

-The core ideas (layer-wise allocation and token merging) are largely combinations of existing techniques. The EMA threshold mechanism is straightforward and lacks theoretical justification.The dynamic allocation strategy is similar to existing attention-based pruning methods

-There is insufficient analysis. There is no ablation study on the impact of different attention patterns across layers. There is also limited discussion on why the 3:1 ratio for N:M works best.

-Regarding experimental concerns, the comparison with baselines is not entirely fair as some methods (like H2O) are designed for different scenarios. There is no discussion on the trade-off between computational overhead and memory savings.

**Questions:**

-How does the computational overhead of D2O scale with increasing sequence lengths and model sizes?

-What are the trade-offs between the compression ratio and the quality of the generated output?

-How does D2O handle extreme cases, such as sequences longer than 100k tokens?

-Why is the exponential function chosen for the allocation strategy? Have other functions been considered?

---

> ### Author Response · Authors · 2024-11-19
> **Response to Reviewer LdwY (Part 1)**
>
> We would like to thank you for taking the time to review our work. Your questions and comments are precious. We value each of them and have provided detailed responses below:
>
> > **W1: The peculiarity of our core ideas**
>
> Thank you for your comments. We would like to clarify that our method is not a simple combination of existing techniques and introduces several key innovations:
>
> + **Differences from Existing Layer-Wise Allocation and Merging Strategies**:
>
> **Layer-Wise KV Cache Allocation** ：Our attention-variance-based dynamic KV cache strategy differs from heuristic approaches like PyramidKV, which reduce cache size  with increasing layers. PyramidKV cannot dynamically adjust KV cache size based on the specific data sample. In contrast, D2O leverages attention density from the input data to adaptively and accurately allocate KV cache size according to the specific needs of each layer.
>
> **Token Merging Strategy**: Unlike previous merging method such as CaM,  which focuses on token-level merging, D2O performs a layer-level operation, dynamically selecting cache size based on attention variance across layers. For merging, CaM uses accumulated attention scores and a Bernoulli mask to evaluate token importance, whereas D2O sorts tokens by attention scores, calculates their similarity, and applies an EMA (Exponential Moving Average) threshold to determine merging. Additionally, CaM primarily merges value caches, while D2O first merges key caches and then applies merging weights to value caches. Our experimental results demonstrate the effectiveness of D2O’s dynamic layer-level and token-level strategies.
>
> [1] PyramidKV: Dynamic KV Cache Compression based on Pyramidal Information Funneling.
>
> [2] CaM: Cache Merging for Memory-efficient LLMs Inference.
>
> + **Heuristic Design of the EMA-Threshold Strategy**:
>
> As shown in Figure 1, attention patterns in higher layers exhibit a staircase-like pattern, focusing on local windows. Only a few critical tokens exist outside these windows, and indiscriminate merging can introduce redundant information or noise, affecting inference accuracy. Our EMA-threshold strategy dynamically selects essential tokens based on previous and current similarity, preserving critical information and reducing redundancy.
>
> + **Distinction from Traditional Attention-Based Pruning:**
>
> While attention-based pruning removes redundant representation at the intra-layer level, our dynamic allocation strategy operates at the Inter-layer level, using attention variance to allocate KV cache size across layers. By implementing an inverse-softmax-based dynamic allocation, we achieve superior results over uniform allocation, as shown in our ablation studies. The theoretical analysis in Appendix A.9 and experiment results in the paper further supports the effectiveness of our layer-level allocation over token-level pruning approaches.
>
> > **W2: Explanation on insufficient analysis**
>
> Thank you for your comments.
>
> 1. We provide a visualization of attention patterns across different datasets and models in Figure 1 and Appendix A.8 ("Visualization of Attention Weights Across Various Datasets"), offering insights into how attention varies by layer and input data.
>
> 2. Regarding the choice of the 3:1 ratio for N:M we include an ablation study in the appendix A.3 (More details of hyper-parameters determination). Additionally, the attention pattern visualizations in Figure 1 and Appendix A.8 illustrate that allocating a smaller cache size to the immediate neighboring window and reserving more capacity for distant contexts allows D2O to retain critical information from longer contexts more effectively.
>
>
> > **W3: Explanation on experimental fairness**
>
> Thank you for pointing out these concerns.
>
> 1. Regarding the comparison with baselines, we ensured fairness by using the default parameter settings provided in their respective papers and GitHub repositories.  **Since all our experimental scenarios involve generative evaluation**, we only use the **official generative-style** code from H2O as a baseline for comparison.
> For D2O, we applied a unified set of hyperparameters derived from a hyperparameter search on a subset of datasets, as detailed in Appendix A.3. Furthermore, all methods were compared under the same compression rate to ensure consistency in evaluating performance.
>
> 2. For the trade-off between computational overhead and memory savings, we conducted detailed analyses in Section 5.5 and Appendix A.5. In Section 5.5, we evaluated the throughput of our method under varying batch sizes and sequence lengths on an A100 80G GPU. Additionally, Appendix A.5 provides a computational cost analysis, comparing prompt encoding and decoding latencies across different sequence lengths.
> These analyses validate both the computational efficiency and memory-saving capabilities of our approach, and we appreciate your suggestion to highlight these aspects.

---

> ### Author Response · Authors · 2024-11-19
> **Response to Reviewer LdwY (Part 2)**
>
> > **Q1: The growth of computational overhead**
>
> Thank you for this question. We present results on D2O’s computational overhead with increasing sequence lengths in Appendix A.5 ("Computational Cost Analysis"). In response to your suggestion, we have now included additional experiments comparing latency across different model sizes, specifically Meta-Llama-3-8B, Llama-2-13B, Code-Llama-34B, and Meta-Llama-3-70B, under varying sequence lengths for encoding/prompt encoding. These experiments follow the setup in Table 12 of Appendix A.5.
> For these tests, we used a single A100 80G GPU for the 8B and 13B models, two GPUs for the 34B model, and four GPUs for the 70B model. As shown in the results, inference time scales with both sequence length and model size, with larger models experiencing increased latency due to computational complexity and communication overhead, even with a fixed-size KV cache optimization.
>
> #### Overall Generation Duration (s). （KV cache size=256）
> | prompt Len + Decoding Len | 8B | 13B | 34B | 70B |
> |---------------------------|----------|---------|---------|----------|
> | 256 + 512 | 29.454 | 53.02 | 147.27 | 279.81 |
> | 512 + 1024 | 58.528 | 105.35 | 292.64 | 556.02 |
> | 1024 + 2048 | 121.191 | 218.14 | 585.36 | 1151.31 |
> | 2048 + 4096 | 232.398 | 418.32 | 998.79 | 2015.78 |
>
> > **Q2: The trade-off between the compression ratio and the output quality**
>
> Thank you for the question. We have analyzed the trade-offs between compression ratio and generation quality in Section 5.2 ("Comparative Analysis of KV Cache Compression Ratios") and Figure 5. Our results show that D2O, through its token-level dynamic merging and layer-level KV cache allocation, effectively preserves essential context across varying compression ratios, thereby enhancing reasoning accuracy without compromising performance. This balance highlights D2O’s ability to maintain output quality even as compression increases.
>
> > **Q3 Dealing with extreme cases**
>
> Thank you for raising this important question. To evaluate D2O's performance on sequences longer than 100K tokens, we conducted additional experiments using the InfiniteBench dataset (average sequence length >100K). These experiments utilized the Llama-3.1-8B-Instruct (128K) model and compared D2O with representative baselines, including eviction-based methods (H2O), layer-wise KV cache reduction (PyramidKV), and value token merging (CaM). All tests were performed on four A100 GPUs with 80GB of memory.
> The results, presented in the table below, show that D2O consistently outperforms the baselines, even for sequences exceeding 100K tokens. This demonstrates that the combination of D2O's dynamic token-level merging and dynamic layer-wise cache allocation effectively preserves context and ensures robust performance in extreme long-context scenarios.
>
>
> #### Infinite-Bench (Llama-3.1-8B-Instruct-128K) / (20% KV cache)
> |             | Passkey | Number_string | Kv_retrieval | Lookbook_choice_eng | Lookbook_qa_eng | Math.F  | code_debug |
> |-------------|---------|---------------|--------------|---------------------|------------------|---------|------------|
> | Full_Cache  | 100     | 99.52         | 28.66        | 62.15               | 25.78            | 35.86   | 26.94      |
> | H2O         | 62.15   | 63.84         | 18.85        | 51.42               | 16.13            | 27.65   | 19.35      |
> | PyramidKV   | 65.25   | 73.88         | 19.82        | 52.91               | 19.36            | 28.19   | 24.08      |
> | CaM         | 76.95   | 78.55         | 23.99        | 51.03               | 20.72            | 32.07   | 23.59      |
> | D2O         | **86.75**   | **91.35**         | **25.88**        | **54.35**               | **24.84**            | **33.85**   | **25.54**      |
>
> > **Q4 Allocation strategy based on exponential function**
>
> Thank you for the question. In Section 5.4, we compare Exponential Decay, Uniform Allocation, and our proposed Reciprocal of Variance. The ablation study shows that the inverse variance softmax effectively allocates KV cache based on attention variance, leading to superior performance across key metrics, demonstrating its adaptability and efficiency.
>
> You have offered constructive and valuable suggestions, making our submission more solid and complete. We promise to include the above results and analyses in the final version. Once again, we sincerely express our best gratitude for your effort and time!
>
> Best wishes,
>
> All authors of Submission 5363.

---

> ### Author Response · Authors · 2024-12-02
> **Follow-Up on Rebuttal Discussion for Submission #5363**
>
> Dear Reviewer LdwY,
>
> We hope this message finds you well.
>
> We have addressed the comments and concerns raised in the reviews to the best of our ability and are keen to receive any further feedback or clarification. **With the decision deadline approaching**, we would like to kindly check if there are any remaining questions or points you would like us to address.
>
> We greatly appreciate the time and effort you and the other reviewers have dedicated to this process, and we are happy to provide any additional information if needed.
>
> Thank you for your attention, and we look forward to hearing from you soon.
>
> Best regards,
>
> Authors of Submission #5363

---

> > ### Author Response · Authors · 2024-12-03
> >
> > Dear Reviewer LdwY,
> >
> > Thank you for your valuable feedback on our paper. We have carefully addressed all comments in the rebuttal and believe the revisions have improved the paper significantly.
> >
> > As the rebuttal phase is nearing its end, we kindly ask if you would consider improving the scores, should our clarifications meet your expectations.
> >
> > We greatly appreciate your time and effort. Please let us know if further clarification is needed.
> >
> > Best regards,
> > Authors

---

### Official Review · Reviewer_JAQq · 2024-11-04

**Soundness:** 4
**Presentation:** 3
**Contribution:** 3
**Rating:** 6
**Confidence:** 4

**Summary:**

This paper presents Dynamic Discriminative Operations D2O, an efficient generation method in large language models. D2O uses layer-level to dynamically adjust cache usage and token-level strategies to merge discarded tokens. D2O achieves significant memory savings and improves throughput without sacrificing generation quality, as demonstrated in various benchmarks and LLM architectures.

**Strengths:**

- D2O takes into account the phenomenon of varying attention score densities across different layers in LLMs, dynamically allocating KV cache budgets for each layer. Additionally, for important discarded tokens, D2O uses a weighted merging approach to retain their information, mitigating information loss during token eviction.
- Compared to various KV cache compression methods, D2O demonstrates consistent superiority across a wide range of benchmarks and different models. With a comparable improvement in throughput, D2O achieves significant performance gains in task execution.
- This paper provides extensive ablation analysis, validating the effectiveness of each component within D2O.

**Weaknesses:**

- The Introduction states that "existing methods equally treat all the layers and indiscriminately evict KV pairs at each layer." However, some prior work has already considered allocating different KV cache budgets across layers, such as PyramidInfer[1] and PyramidKV[2]. The allocation method in D2O should be compared with those in previous works.
- The use of mathematical symbols is confusing. For example, the F representing variance in line 207 is a scalar and should not be in bold, as well as u_ij in line 261.
- The experiment setup is not clear enough. For example, in Section 5.3, the maximum length setting for the LongBench test is mentioned.

[1] PyramidInfer: Pyramid KV Cache Compression for High-throughput LLM Inference (Yang et, al.)

[2] PyramidKV: Dynamic KV Cache Compression based on Pyramidal Information Funneling (Cai et, al.)

**Questions:**

- As the generation length increases, if a certain KV Cache budget is maintained, the effect of token merging is expected to weaken. Are there any experiments on this?
- The performance testing in the paper separates the benchmarks from the throughput tests. In the LongBench, the prompts are usually long while the outputs are short, so the methods presented in this paper may not save much time on such tasks. Are there more suitable benchmarks where the generated text is longer, allowing for simultaneous evaluation of generation speed and performance?

---

> ### Author Response · Authors · 2024-11-19
> **Reponse to Reviewer JAQq**
>
> We sincerely appreciate your review. Your effort has ensured that our submission received adequate attention and thorough review, and also provided us with lots of suggestions . We value each of your suggestions and provide the following responses:
>
> > **W1: Comparing the allocation method in D2O with other works**
>
> Thank you for highlighting this point. In our view, those works,  such as PyramidInfer [1] and PyramidKV [2], primarily rely on heuristically designed functions to allocate KV cache budgets based on the observed trend that attention density decreases with increasing layers. While effective to some extent, their limitation lies in the lack of adaptability to individual data samples, as these methods do not incorporate prior knowledge specific to the input data.
> In contrast, our proposed D2O employs a dynamic KV cache allocation mechanism that leverages the variance of attention scores across different layers for each specific input. This enables a more adaptive and data-driven allocation of KV cache, setting D2O apart from the heuristic approaches used in prior works. We believe this distinction underscores the novelty and practical advantages of our method.
> As you suggest, we added additional comparison experiments. The results show that D2O outperforms other non-adaptive KV cache allocation strategies in most of the LongBench datasets. This improvement highlights the advantage of our dynamic allocation strategy, which utilizes an inverse variance softmax function to adjust the KV cache size in each layer. And our adaptive approach is particularly effective in long-context scenarios, where maintaining relevant information across layers is crucial.
> #### LongBench (Mistral-7B-Instruct-v0.2) / (20% KV Cache)
> | Model         | Qasper | QMSum | MultiNews | TREC  | TriviaQA | SAMSum | LCC   | RepoBench-P |
> |---------------|--------|-------|-----------|-------|----------|--------|-------|-------------|
> | Full_cache    | 29.80   | 24.44 | 26.28     | 66.67 | 86.16    | 41.11  | 56.91 | 49.09        |
> | PyramidKV     | 23.80   | 20.6  | **24.22**     | 59.44 | 82.90    | 35.31  | 53.52 | 42.11        |
> | PyramidInfer  | 23.38  | 19.8  | 21.23     | 58.35 | 78.24    | 35.18  | 52.35 | 41.88        |
> | D2O           | **25.72**  | **21.9**  | 24.06     |**62.99** | **84.02**    | **38.03**  | **55.17** | **46.15**        |
>
> We appreciate your suggestion and have incorporated citations to PyramidInfer [1] and PyramidKV [2] in the Related Work section of the revised version of our paper.
>
> > **W2 Confusing mathematical symbols and typos**
>
> Thank you for pointing this out. We acknowledge that these are typographical errors in the paper and will correct them by adjusting the mathematical symbols accordingly. In the revised version, we will make the necessary modifications.
>
> > **W3 The clearer experiment setup**
>
> Thank you for the suggestion. In the revised version, we will include detailed length information for each dataset in LongBench. Specifically, the length of each dataset from  LongBench can be categorized into three ranges: under 4K, 4-8K, and over 8K tokens. The average lengths are as follows: NrtvQA (18409), Qasper (3619), MF-en (4559), HotpotQA (9151), 2WikiMQA (4887), Musique (11214), GovReport (8734), QMSum (10614), MultiNews (2113), TREC (5177), TriviaQA (8209), SAMSum (6258), PCount (11141), PRe (9289), LCC (1235), and RepoBench-P (4206).
>
> > **Q1 Experiment on the effect of generation length and token merging**
>
> Thank you for this question. In Appendix A.4 (Generated Samples of Multi-turn Conversations), our MT-bench experiments show that D2O’s dynamic merging strategy outperforms non-merging eviction strategies like H2O and StreamLLM, even with a fixed KV cache budget. These results indicate that D2O effectively mitigates the impact of increased generation length, preserving essential information more efficiently.

---

> ### Author Response · Authors · 2024-11-19
> **Reponse to Reviewer JAQq (Part 2)**
>
> > **Q2 The benchmark for evaluating both generation speed and performance**
>
> Thank you for the constructive suggestion. To assess inference time and generation quality during long-generation tasks, we conducted additional tests on the longbook_sum_eng dataset from InfiniteBench [1], which has an average output length of 1.1K tokens. Due to computational constraints, we tested the first 20 samples using the Llama-3.1-8B-Instruct (128K) model [2], comparing D2O with representative baselines, including the eviction-based method (H2O), layer-wise KV cache reduction (PyramidKV), and value token merge (CaM). The experiment was conducted using four A100 GPUs with 80GB.
>
> #### Lookbook_sum_eng (Llama-3.1-8B-Instruct ) / (20% KV Cache)
> | | rouge_lsum (f1) | inference time (min) |
> |------------|------------------|----------------------|
> | Full Cache | 36.87 | 259.64 |
> | H2O | 28.12 | 190.91 |
> | PyramidKV | 31.92 | 206.06 |
>  | CaM | 33.82 | 181.51 |
> | D2O | 35.48 | 201.27 |
>
> As shown in the table, D2O achieved the best performance, with total inference time between H2O and PyramidKV and minimal latency differences.
>
> [1] https://github.com/OpenBMB/InfiniteBench
>
> [2] https://huggingface.co/meta-llama/Llama-3.1-8B-Instruct
>
> You have offered constructive and valuable suggestions, making our submission more solid and complete. Once again, we sincerely express our best gratitude for your effort and time!
>
> Best wishes,
>
> All authors of Submission 5363.

---

> ### Author Response · Authors · 2024-11-30
> **Looking Forward to Your Feedback**
>
> Dear Reviewer JAQq,
>
> We greatly appreciate your valuable comments and suggestions on our paper. We have made every effort to address your concerns and sincerely hope these updates align with your expectations. In response, we have carefully revised our work and added additional experimental validations as necessary. We look forward to receiving your feedback at your earliest convenience.
>
> Wishing you a wonderful Thanksgiving!
>
> Best regards,
>
> Team 5363

---

> ### Author Response · Authors · 2024-12-03
>
> Dear Reviewer JAQq,
>
> Thank you for your valuable feedback on our paper. We have carefully addressed all comments in the rebuttal and believe the revisions have improved the paper significantly.
>
> As the rebuttal phase is nearing its end, we kindly ask if you would consider improving the scores, should our clarifications meet your expectations.
>
> We greatly appreciate your time and effort. Please let us know if further clarification is needed.
>
> Best regards, Authors

---

### Author Response · Authors · 2024-11-21
**Response Summary and Paper Revision.**

Dear Reviewers,

We sincerely thank you for their valuable feedback, which has significantly improved the quality and rigor of our paper. Based on your
suggestions, we have revised and updated the paper, highlighting changes and additions in **blue** within the revised PDF. Below, we summarize the key revisions made:

- **1. Addition of Baselines:** In response to comments from reviewers JAQq, VHpa, and FB7s, we have incorporated comparisons with the latest baselines, including SnapKV [1], PyramidKV [2], PyramidInfer [3], SirLLM [4], and LoCoco [5]. These results are presented in the updated Appendix A.10 (highlighted in blue) and **Table 18**.

- **2. Expanded Related Work:** Following suggestions from reviewers aSVX, FB7s, and JAQq, we have added descriptions of additional relevant works, including NaCL [6], SirLLM [4], PyramidKV [2], and PyramidInfer [3], in the Related Work section, now highlighted in blue in the main text.

- **3. Generative Long-Context Evaluation:** Per reviewer JAQq's feedback, we conducted experiments to evaluate both generation speed and performance on generative long-context tasks. These results are included in **Appendix A5 (lines 928–937)** and **Table 15**.

- **4. Inclusion of InfiniteBench and Ruler Results:** As recommended by reviewers LdwY and VHpa, we added evaluations using InfiniteBench [7] and Ruler [8]. The results are detailed in **Appendix A.11** and presented in **Tables 19 and 20**.

- **5. Computational Overhead Analysis:** Addressing reviewer LdwY's suggestion, we evaluated the growth of computational overhead with increasing model sizes and context lengths. These results are included in **Appendix A5 (lines 921–927)** and **Table 14**.

- **6. Needle-in-a-Haystack Results:** Based on reviewer aSVX’s feedback, we expanded the analysis of needle-in-a-haystack results with corresponding inference times, now detailed in **Appendix A5 (lines 934–937)** and **Table 16**.

- **7. Clarity and Typos:** To address comments from reviewers FB7s and JAQq, we corrected and clarified certain expressions and typos. For example: **Line 181-182**: "uniform eviction strategy" was revised to "uniform layer-wise size."  **Line 207 and 261**: Removed bold formatting for scalar variables. These corrections are highlighted in blue in the revised text.

Once again, we deeply appreciate the time and effort you have dedicated to improving this paper. Your insights have made the paper stronger and more comprehensive. Wishing you a Merry Christmas and a Happy New Year!

Bests,

Your friends,

Team of 5363

**References:**
[1] Li, Yuhong et al. “SnapKV: LLM Knows What You are Looking for Before Generation.” ArXiv abs/2404.14469 (2024).

[2] Cai, Zefan et al. “PyramidKV: Dynamic KV Cache Compression based on Pyramidal Information Funneling.” ArXiv abs/2406.02069 (2024).

[3] Yang, Dongjie et al. “PyramidInfer: Pyramid KV Cache Compression for High-throughput LLM Inference.” Annual Meeting of the Association for Computational Linguistics (2024).

[4] Yao, Yao et al. “SirLLM: Streaming Infinite Retentive LLM.” ArXiv abs/2405.12528 (2024).

[5] Cai, Ruisi et al. “LoCoCo: Dropping In Convolutions for Long Context Compression.” ArXiv abs/2406.05317 (2024).

[6] Chen, Yilong et al. “NACL: A General and Effective KV Cache Eviction Framework for LLM at Inference Time.” ArXiv abs/2408.03675 (2024).

[7] Zhang, Xinrong et al. “∞Bench: Extending Long Context Evaluation Beyond 100K Tokens.” ArXiv abs/2402.13718 (2024).

[8] Hsieh, Cheng-Ping et al. “RULER: What's the Real Context Size of Your Long-Context Language Models?” ArXiv abs/2404.06654 (2024).

---

### Meta-Review · Area_Chair_DjTu · 2024-12-24

**Metareview:**

This paper introduces a method for efficient LLM inference, D2O, through dynamic KV cache optimization. The key contributions are (1) layer-wise cache allocation based on attention density patterns, and (2) a token-level strategy combining eviction with dynamic merging using the EMA threshold.

Reviewers respect the conributions such as task importance, effective apporach, and promising results with extensive experiments while a reviewer raised limited novelty issue.

AC carefully read the paper, reviewer comments, and author response. It seems that the authors provided clear explanation and additional experiments on the issues raised by LdwY.

Considering the contributions and no additional feedback, AC recommends accepting this paper.

**Additional Comments On Reviewer Discussion:**

The initial scores were two 5 and three 6.

Durint the rebuttal period, FB7s raised his/her score as 6. considering the additional experiments including LongBench by the authors.

So, the final scores are four 6 and one 5. Because LdwY did not gave any comments on the authors' response and AC-reviewer disccusion, AC confirmed the authors response, which seems to address the main concerns raised by LdwY.

---

### Decision · Program_Chairs · 2025-01-22

Accept (Poster)